# Mitochondria-adaptor TRAK1 promotes kinesin-1 driven transport in crowded environments

Verena Henrichs [1,2], Lenka Grycova[1], Cyril Barinka [1], Zuzana Nahacka[1], Jiri Neuzil[1,3], Stefan Diez [4,5], Jakub Rohlena [1], Marcus Braun [1✉] & Zdenek Lansky [1✉]

Intracellular trafficking of organelles, driven by kinesin-1 stepping along microtubules, underpins essential cellular processes. In absence of other proteins on the microtubule surface, kinesin-1 performs micron-long runs. Under crowding conditions, however, kinesin-1 motility is drastically impeded. It is thus unclear how kinesin-1 acts as an efficient transporter in intracellular environments. Here, we demonstrate that TRAK1 (Milton), an adaptor protein essential for mitochondrial trafficking, activates kinesin-1 and increases robustness of kinesin-1 stepping on crowded microtubule surfaces. Interaction with TRAK1 i) facilitates kinesin-1 navigation around obstacles, ii) increases the probability of kinesin-1 passing through cohesive islands of tau and iii) increases the run length of kinesin-1 in cell lysate. We explain the enhanced motility by the observed direct interaction of TRAK1 with microtubules, providing an additional anchor for the kinesin-1-TRAK1 complex. Furthermore, TRAK1 enables mitochondrial transport in vitro. We propose adaptor-mediated tethering as a mechanism regulating kinesin-1 motility in various cellular environments.

[1] Institute of Biotechnology of the Czech Academy of Sciences, BIOCEV, Vestec, 25250 Prague West, Czech Republic. [2] Faculty of Science, Charles University in Prague, 12800 Prague, Czech Republic. [3] School of Medical Science, Griffith University, Southport, QLD 4222, Australia. [4] B CUBE - Center of Molecular Bioengineering and Cluster of Excellence Physics of Life, Technische Universität Dresden, 01307 Dresden, Sachsen, Germany. [5] Max Planck Institute of Molecular Cell Biology and Genetics, 01307 Dresden, Sachsen, Germany. ✉email: marcus.braun@ibt.cas.cz; zdenek.lansky@ibt.cas.cz

I n eukaryotic cells, microtubules constitute a major part of the cytoskeleton and provide a multi-functional scaffold crucial for intracellular long-range transport. Microtubule-based transport is particularly important in neurons, where it enables an efficient distribution of cargo, such as mitochondria, along elongated axons to distal regions of the cell. Dysfunctions in the distribution of mitochondria are connected with neurodegenerative diseases, including Alzheimer's disease, Huntington's disease or amyotrophic lateral sclerosis[1–3]. Active mitochondrial transport is equally essential for the redistribution of mitochondria during mitosis[4,5] and trafficking of mitochondria between cells through tunnelling nanotubes[6–8] is relevant in tumour initiation and progression[7,9,10].

Trafficking of cargo, including mitochondria, is driven by molecular motors such as kinesin-1[11]. By hydrolysing ATP in its N-terminal motor domains, kinesin-1 heavy chain (further referred to as kinesin-1) moves in steps towards the plus-end of microtubules. Thus, kinesin-1 drives anterograde axonal transport[12–14] of mitochondria, which in vivo are transported in bursts of motion, covering distances of tens of micrometres[15]. Kinesin-1 is moderately processive in absence of other proteins on the microtubule surface in vitro, meaning that it can perform about 100 consecutive steps towards the microtubule plus-end, covering hundreds of nanometres before dissociating from the microtubule[16]. Reconstitutions of kinesin-1 stepping demonstrate that the coupling of multiple molecular motors to a single cargo increases cargo processivity[17–20]. Indeed, in living cells, various cellular cargoes are transported by multiple molecular motors[21–23]. In the cytoplasm of living cells, however, microtubules are heavily decorated by a large variety of proteins, crowding the microtubule surface[24]. In vitro experiments demonstrate that crowding strongly impedes kinesin-1-driven transport through a drastic reduction of kinesin-1 processivity[25–28]. One of the key regulators of microtubule-based trafficking in neurons, the intrinsically disordered protein tau[29–31], can form densely crowded cohesive islands on microtubules, which strongly impede kinesin-1 motility[32–34]. To enable robust long-range kinesin-1-driven transport in cells, additional mechanisms, complementary to the coupling of multiple molecular motors, are thus likely required to overcome the hindering effect of crowding on the microtubule surface.

Binding of cargo, such as mitochondria, is mediated by adaptor proteins interacting with the C-terminal tail domain of kinesin-1. In absence of cargo, kinesin-1 is auto-inhibited via an interaction of the cargo-binding domain with the motor domain[35–37]. Mitochondria as cargo are physically linked to kinesin-1 by the adaptor proteins Miro and Milton[38–41]. Two mammalian Milton homologues are known, TRAK1 and TRAK2, (trafficking kinesin protein 1 and 2, respectively), with TRAK1 being the preferred binding partner of kinesin-1[42,43]. Knockdown of TRAK1 results in neurodegeneration, which, intriguingly, is suppressed by additional knockdown of tau[44]. Binding of TRAK1 to kinesin-1 is mediated by a direct association of the coiled-coil domain of TRAK1 with the regulatory tail domain of kinesin-1[39,41–43,45,46]. We thus wondered whether the adaptor protein TRAK1 regulates the motility of kinesin-1, and specifically, whether this regulation affects kinesin-1 motility in crowded environments, such as in the cohesive tau islands.

We report that TRAK1 activates kinesin-1, increases the molecular motor's processivity and thus promotes long-range transport on densely crowded microtubules and within cohesive tau islands. We explain these observations by TRAK1-mediated anchoring of kinesin-1 to the microtubule surface and we propose auxiliary anchoring, mediated by adaptor proteins, as a mechanism regulating kinesin-1 transport, which is particularly supportive in crowded environments.

## Results

**TRAK1 activates KIF5B.** Full-length KIF5B (herein referred to as KIF5B; Supplementary Fig. 1a) is a kinesin-1 heavy chain encoded in the human genome. It is auto-inhibited in absence of a cargo[35], unable to interact with microtubules. In order to study the interaction of KIF5B with TRAK1, we immobilized microtubules onto a coverslip surface and added 6 nM GFP-labelled full-length KIF5B (KIF5B-GFP; Supplementary Fig. 1a) and/or 175 nM mCherry-labelled full-length TRAK1 (mCherry-TRAK1; Supplementary Fig. 1a, b). To visualize molecular interactions with microtubules we used total internal reflection fluorescence (TIRF) microscopy ("Methods"). Imaging of KIF5B-GFP in the absence of TRAK1 showed that it indeed was auto-inhibited, unable to bind to microtubules (Fig. 1a, b). Consistent with previously published data[35,47,48], we observed occasional brief diffusion of KIF5B-GFP along microtubules and sporadic processive runs occurring with a landing rate of $1.31 \pm 0.51$ molecules mm$^{-1}$ s$^{-1}$ (mean ± standard deviation, $n = 87$ molecules; Fig. 1a). When we next imaged mCherry-TRAK1 in the absence of KIF5B-GFP, we unexpectedly observed single molecules of mCherry-TRAK1 diffusing along microtubules (Fig. 1c), revealing that TRAK1 contains a microtubule-binding domain. The C-terminal region of TRAK1 exhibits a high isoelectric point of 10, and is thus likely to interact electrostatically with the negatively charged microtubule surface. The mCherry-labelled TRAK1 mutant lacking this region (amino acids 636-953; mCherry-TRAK1Δ; Supplementary Fig. 1a, c) indeed did not show any interaction with microtubules (Fig. 1d) confirming that this region is the microtubule-binding domain of TRAK1. When we finally imaged KIF5B-GFP in presence of mCherry-TRAK1 (Fig. 1e) we observed directed processive motility, detected in the GFP and the mCherry channel (82% and 4%, respectively), with 14% of the events colocalizing in both channels (Fig. 1a), showing that TRAK1 and KIF5B can form a processive complex. We note that in presence of KIF5B-GFP, we did not observe any diffusion of mCherry-TRAK1. The overall landing rate increased to $4.49 \pm 2.33$ molecules mm$^{-1}$ s$^{-1}$ (mean ± standard deviation, $n = 547$ molecules; Fig. 1a), indicating that TRAK1 activates KIF5B. To elucidate whether the interaction of TRAK1 with microtubules is necessary for the activation of KIF5B, we imaged KIF5B-GFP in presence of 175 nM mCherry-TRAK1Δ. The deletion of the C-terminus of mCherry-labelled TRAK1 did not compromise its binding to KIF5B. Similarly, as when in presence of the full-length mCherry-TRAK1, we observed directed processive motility in both channels (Fig. 1f) and an overall landing rate of $5.85 \pm 1.28$ molecules mm$^{-1}$ s$^{-1}$ (mean ± standard deviation, $n = 308$ molecules; Fig. 1a) indicating that TRAK1-binding to microtubules is dispensable for the activation of KIF5B.

**TRAK1 increases KIF5B processivity.** We next determined the run length, the interaction time with microtubules and the velocity of the processive KIF5B-TRAK1 transport complexes observed in experiments in Fig. 1. We analysed only traces of mCherry-TRAK1 (which by itself does not perform directed processive runs; Fig. 1c) to exclude KIF5B-GFP molecules that were not in complex with mCherry-TRAK1. Interestingly, we observed that the median run length of the KIF5B-GFP-mCherry-TRAK1 complex of 8.14 μm (95% confidence interval, CI$_{95}$ (4.57, 15.56) μm, $n = 189$ molecules) was significantly higher than the median run length of the truncated KIF5B-GFP-mCherry-TRAK1Δ complex of 3.49 μm (CI$_{95}$ (2.88, 4.53) μm, $n = 282$ molecules) (Fig. 2a). A similar trend was observed for the interaction time of the complexes with microtubules, with the median interaction time of 19.27 s (CI$_{95}$ (11.48,-) s, $n = 189$ molecules) for the KIF5B-GFP-mCherry-TRAK1 complex and 5.33 s (CI$_{95}$ (3.69, 8) s, $n = 282$

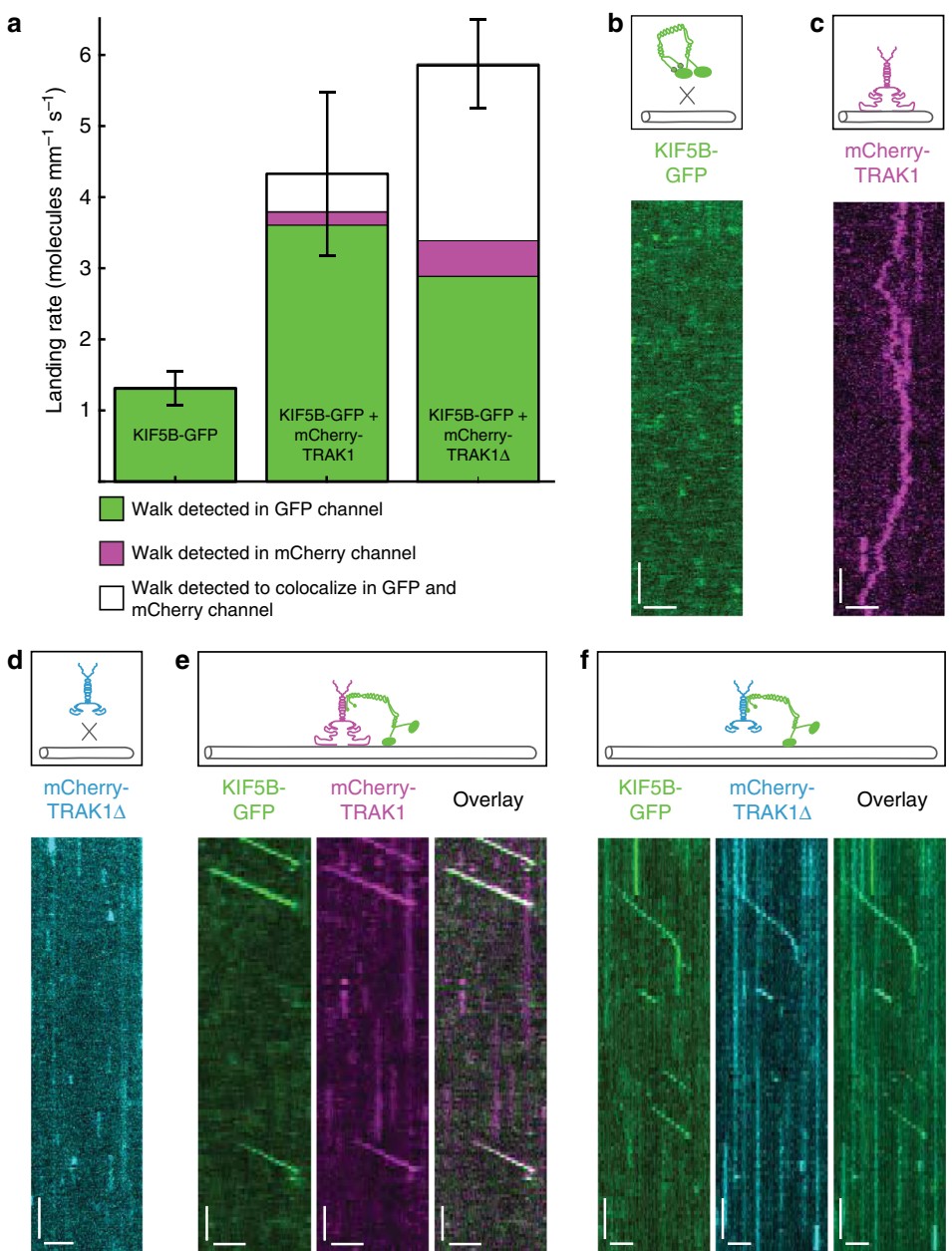

**Fig. 1 TRAK1 activates KIF5B. a** Landing rates for processively walking KIF5B-GFP molecules alone (left, $n = 87$ molecules, $N = 4$ experiments), in presence of mCherry-TRAK1 (middle, $n = 547$ molecules, $N = 4$ experiments) and in presence of mCherry-TRAK1Δ (right, $n = 308$ molecules, $N = 4$ experiments), respectively, detected in the GFP channel (green), mCherry channel (magenta) or colocalizing in both channels (white). Data are presented as mean values ± standard deviation. Source data are provided as a Source Data file. Schematic illustrations and kymographs of **b** auto-inhibited KIF5B-GFP not interacting with the microtubule (experiment was repeated 60 times independently with similar results), **c** mCherry-TRAK1 diffusing along the microtubule (experiment was repeated 28 times independently with similar results), **d** mCherry-TRAK1Δ not interacting with the microtubule (experiment was repeated four times independently with similar results), **e** KIF5B-GFP (green) colocalizing with mCherry-TRAK1 (magenta) and moving processively along the microtubule and **f** KIF5B-GFP colocalizing with mCherry-TRAK1Δ (cyan) and moving processively along the microtubule. Horizontal scale bars 2 μm, vertical 5 s.

molecules) for the KIF5B-GFP-mCherry-TRAK1Δ complex (Fig. 2b), while the average velocity of the KIF5B-GFP-mCherry-TRAK1 complex of $753 \pm 347$ nm s$^{-1}$ (mean ± standard deviation, $n = 189$ molecules) was slightly lower than the average velocity of the KIF5B-GFP-mCherry-TRAK1Δ complex of $969 \pm 506$ nm s$^{-1}$ (mean ± standard deviation, $n = 282$ molecules) (Fig. 2c). These results indicate that the interaction of TRAK1 with the microtubule increases the processivity of the KIF5B-TRAK1 complex.

To directly investigate how TRAK1 modulates KIF5B motility, we generated a constitutively active KIF5B construct by removing

the KIF5B inhibitory domain (amino acids 906-963, KIF5BΔ; Supplementary Fig. 1a). Visualisation of GFP-labelled KIF5BΔ (KIF5BΔ-GFP; Supplementary Fig. 1a) interacting with microtubules confirmed that it moved processively (Supplementary Fig. 2a, green). Next, we imaged the interaction of KIF5BΔ-GFP with mCherry-TRAK1 on microtubules (Supplementary Fig. 2a, magenta). Similar to full-length KIF5B-GFP, we observed colocalization of KIF5BΔ-GFP and mCherry-TRAK1 during processive runs (Supplementary Fig. 2b), showing that removal of the inhibitory domain of KIF5B did not disrupt the KIF5B

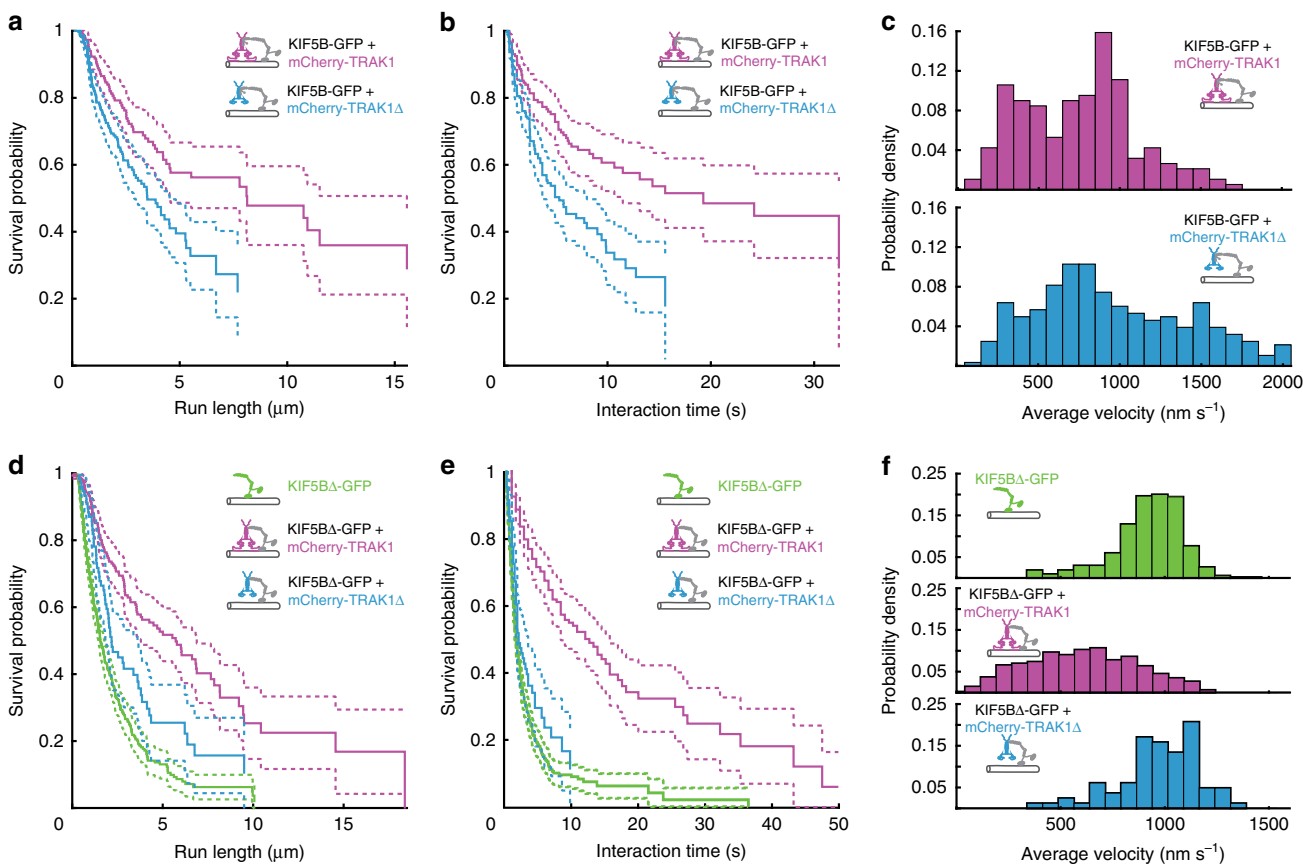

**Fig. 2 TRAK1 increases KIF5B processivity. a, b** Survival probability (Kaplan–Meier estimation) of the run length and interaction time, respectively, of KIF5B-GFP in presence of mCherry-TRAK1 (magenta, $n = 189$ molecules, $N = 3$ experiments) and mCherry-TRAK1Δ (cyan, $n = 282$ molecules, $N = 3$ experiments) ($p = 0.0003$ and $p < 10^{-4}$, respectively). **c** Histograms of the average velocities of KIF5B-GFP in presence of mCherry-TRAK1 (magenta, $n = 189$ molecules, $N = 3$ experiments) and mCherry-TRAK1Δ (cyan, $n = 282$ molecules, $N = 3$ experiments), respectively. **d, e** Survival probability of the run length and interaction time, respectively, of KIF5BΔ-GFP (green, $n = 499$ and 534 molecules, respectively, $N = 2$ experiments), KIF5BΔ-GFP in complex with mCherry-TRAK1 (magenta, $n = 222$ and 221 molecules, respectively, $N = 4$ experiments) and KIF5BΔ-GFP in complex with mCherry-TRAK1Δ (cyan, $n = 82$, $N = 2$ experiments). In presence of mCherry-TRAK1, the median run length and interaction time of KIF5BΔ-GFP increased (both $p$-values $< 10^{-4}$). The increase was significantly less pronounced in presence of mCherry-TRAK1Δ ($p = 0.0008$ and $p = 0.0418$, respectively). **f** Histograms of the average velocities of KIF5BΔ-GFP (green, $n = 534$ molecules, $N = 2$ experiments), KIF5BΔ-GFP in presence of mCherry-TRAK1 (magenta, $n = 487$ molecules, $N = 4$ experiments) and KIF5BΔ-GFP in presence of mCherry-TRAK1Δ (cyan, $n = 82$ molecules, $N = 2$ experiments). The velocity of KIF5BΔ-GFP decreased only in presence of mCherry-TRAK1 but not in presence of mCherry-TRAK1Δ. See also Supplementary Fig. 2. All two-tailed $p$-values were obtained by a log-rank test. Source data are provided as a Source data file.

interaction with TRAK1. Analysing the fluorescence intensity distributions furthermore indicated that the transport complex comprises one TRAK1 dimer and one KIF5B-GFP dimer (Supplementary Fig. 2c–h, Supplementary Table 1; "Methods"). Strikingly, we observed a several fold increase in the median run length of KIF5BΔ-GFP in presence of mCherry-TRAK1 from 1.54 μm (CI$_{95}$ (1.43, 1.79) μm, $n = 499$ molecules) to 5.77 μm (CI$_{95}$ (3.78, 6.87) μm, $n = 222$ molecules) (Fig. 2d, green and magenta). The median interaction time of KIF5BΔ-GFP with microtubules also increased several folds from 1.84 s (CI$_{95}$ (1.64, 2.05) s, $n = 534$ molecules) in absence of mCherry-TRAK1 to 12.20 s (CI$_{95}$ (8.54, 15.86) s, $n = 221$ molecules) in its presence (Fig. 2e, green and magenta). The average velocity of KIF5BΔ-GFP decreased in presence of mCherry-TRAK1 from $918 \pm 171$ nm s$^{-1}$ (mean ± standard deviation, $n = 534$ molecules) to $599 \pm 262$ nm s$^{-1}$ (mean ± standard deviation, $n = 487$ molecules; Fig. 2f, magenta and green; "Methods") with an increase in the frequency of transient pauses (Supplementary Fig. 2i). Consistent with our hypothesis that TRAK1Δ does not anchor KIF5B to microtubules, mCherry-TRAK1Δ barely affected the KIF5BΔ-GFP median run length (2.17 μm, CI$_{95}$ (1.88, 3.59) μm, $n = 82$

molecules; Fig. 2d, cyan and Supplementary Fig. 2a, cyan), median interaction time (2.22 s, CI$_{95}$ (1.82, 3.63) s, $n = 82$ molecules; Fig. 2e, cyan) and velocity (average velocity $970 \pm 196$ nm s$^{-1}$, mean ± standard deviation, $n = 82$ molecules; Fig. 2f and Supplementary Fig. 2i, cyan). We thus conclude that TRAK1 tethers KIF5B to the microtubule, increasing the run length and the interaction time of the complex with the microtubule while decreasing the transport velocity.

**TRAK1 promotes KIF5B processivity in cohesive islands of tau**. The intrinsically disordered microtubule-associated protein tau can regulate microtubule-based transport by forming cohesive islands on microtubules, strongly impeding kinesin-1 motility[32,34]. Since we observed that TRAK1 increased the processivity of KIF5B, we wondered whether TRAK1 can extend the movement range of kinesin-1 within tau islands. Initially, to assess the effect of tau on the interaction between TRAK1 and microtubules, we formed tau islands on microtubules using GFP-labelled tau (tau-GFP) and imaged the diffusion of mCherry-TRAK1 along the microtubules. We observed mCherry-TRAK1 landing and diffusion restricted to the microtubule regions not

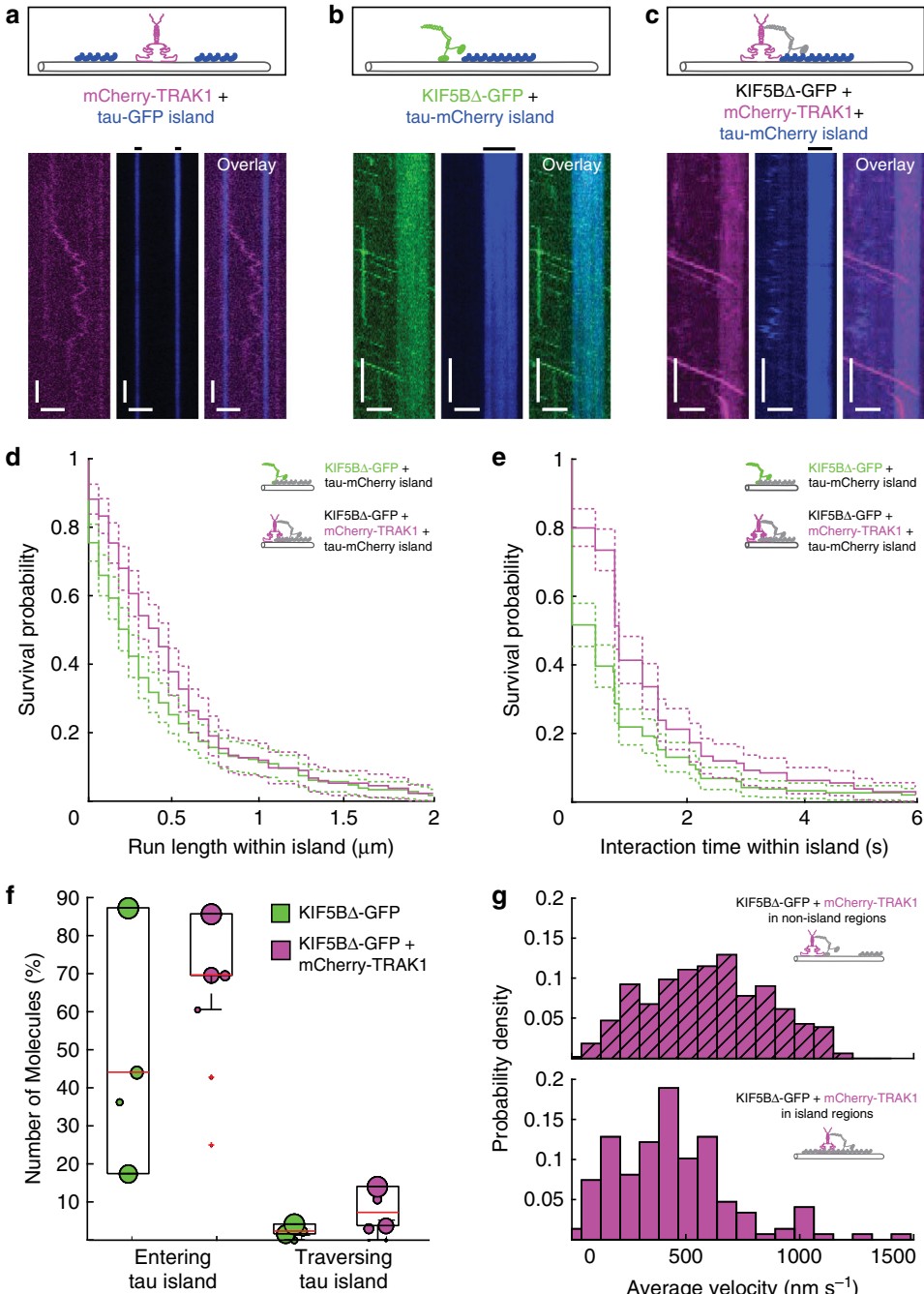

**Fig. 3 TRAK1 promotes KIF5B processivity in cohesive tau islands.** Schematic illustrations and kymographs of **a** mCherry-TRAK1 (magenta) diffusing in non-tau island regions but not entering tau-GFP islands (blue), **b** processive movement of KIF5BΔ-GFP (green) dissociating at the edges of tau-mCherry islands (blue) and **c** KIF5BΔ-GFP in presence of mCherry-TRAK1 (magenta) penetrating into tau-mCherry islands (blue). The positions of the tau-mCherry islands are denoted by black horizontal lines. Horizontal scale bars 1 μm, vertical 5 s. The experiments were repeated 4 times independently with similar results **d, e** Survival probability plots showing the increased median run length and interaction time of KIF5BΔ-GFP within tau-mCherry islands in presence of mCherry-TRAK1 (magenta) in comparison to KIF5BΔ-GFP in absence of mCherry-TRAK1 (green) ($p = 0.0141$ and $p < 10^{-4}$, respectively). Two-tailed $p$-values were obtained by a log-rank test. **f** KIF5BΔ-GFP in presence of mCherry-TRAK1 (magenta, $n = 202$ molecules, $N = 6$ experiments) showed a higher probability to enter and traverse tau-mCherry islands than KIF5BΔ-GFP alone (green, $n = 240$ molecules, $N = 4$ experiments). Red lines present the median values, top and bottom edges of the black boxes present upper and lower quantiles, whiskers present the highest/lowest data points and red crosses present outliers. Data points are weighted by the number of measurements. For island length distributions see Supplementary Fig. 3. **g** Histograms showing the average velocity of KIF5BΔ-GFP-mCherry-TRAK1 complexes in non-island regions (magenta dashed, same dataset as in Fig. 2f) and within tau-mCherry islands (magenta, $n = 148$ molecules, $N = 6$ experiments). Source data are provided as a Source data file.

covered by tau islands (Fig. 3a). We next formed islands using mCherry-labelled tau (tau-mCherry) and investigated its effect on kinesin-1 motility. When KIF5BΔ-GFP encountered a tau-mCherry island, about 56% of the molecules detached directly

at the boundary of the island. The remaining molecules (44% (lower, upper quantile limit 17%, 88%), median ± quantile range, $n = 240$ molecules) penetrated tau-mCherry islands to reach a median run length within the tau island of 0.23 μm (CI$_{95}$ (0.17,

0.23) μm, $n = 240$ molecules) and an interaction time of 0.41 s (CI$_{95}$ (0.00, 0.41) s, $n = 240$ molecules) before detachment (Fig. 3b, d–f). By contrast, complexes of KIF5BΔ-GFP and mCherry-TRAK1 entered tau-mCherry islands in 70% (lower, upper quantile limit 70%, 86%, median ± quantile range, $n = 202$ molecules) of the events and the median run length and interaction time within the islands, when compared to KIF5BΔ-GFP alone, increased about twofold to 0.41 μm (CI$_{95}$ (0.29, 0.46) μm, $n = 202$ molecules) and 0.82 s (CI$_{95}$ (0.75, 0.82) s, $n = 202$ molecules), respectively (Fig. 3c–f). This increase in processivity within the tau-mCherry islands resulted in more than threefold increased probability of KIF5BΔ-GFP (2% (lower, upper quantile limit 2%, 4%, median ± quantile range, $n = 240$ molecules)) to completely traverse a tau-mCherry island when in complex with mCherry-TRAK1 (7% (lower, upper quantile limit 4%, 14%, median ± quantile range, $n = 202$ molecules)) (Fig. 3f, Supplementary Fig. 3). Similar to super-processive kinesin-8 motors stepping through the tau islands[32], the average velocity of KIF5BΔ-GFP in complex with mCherry-TRAK1 decreased from $599 ± 262$ nm s$^{-1}$ (mean ± standard deviation, same dataset as in Fig. 2f) to $442 ± 268$ nm s$^{-1}$ (mean ± standard deviation, $n = 148$ molecules) (Fig. 3g) while traversing the tau-mCherry islands. In combination, these findings demonstrate TRAK1-mediated anchoring of KIF5B enhances the motility of KIF5B-TRAK1 complexes within cohesive tau islands.

**TRAK1 increases KIF5B processivity in crowded environments.** In addition to tau, microtubule surfaces in vivo are crowded with a plethora of other microtubule-associated proteins that hinder the motion of molecular motors. To test the effect of TRAK1-mediated anchoring upon KIF5B encountering fully-blocked binding sites, we first used a GFP-labelled rigor-binding truncated kinesin-1 T93N mutant[25] (Supplementary Fig. 1a; "Methods"). This mutant is unable to hydrolyse ATP[49] and therefore binds with a high affinity to microtubules, forming stationary obstacles that completely mask the KIF5B binding sites. As described before[25], the median run length of KIF5BΔ-mCherry decreased in presence of these static obstacles by ~50% from 1.54 μm (CI$_{95}$ (1.43, 1.79) μm, same dataset as in Fig. 2d) to 0.86 μm (CI$_{95}$ (0.79, 0.98) μm, $n = 547$ molecules). Consistently, the median interaction time of KIF5BΔ-mCherry decreased in presence of the obstacles by about 30% from 1.84 s (CI$_{95}$ (1.64, 2.05) s, same dataset as in Fig. 2e) to 1.40 s (CI$_{95}$ (1.20, 1.43) s, $n = 547$ molecules) (Fig. 4a–c green and red). Addition of mCherry-TRAK1 increased the median run length of KIF5BΔ-mCherry in presence of the obstacles to 1.17 μm (CI$_{95}$ (1.06, 1.29) μm, $n = 1066$ molecules) and restored the median interaction time of KIF5BΔ-mCherry in presence of obstacles to 1.84 s (CI$_{95}$ (1.80, 2.20) s, $n = 1066$ molecules) (Fig. 4a–c magenta). These data demonstrate that TRAK1 promotes KIF5BΔ processivity even when the exact binding sites for the kinesin-1 motor domains are occupied by stationary obstacles.

Secondly, we wondered whether TRAK1-mediated anchoring of KIF5B would suffice to promote stepping in the crowded environment similar to the cytoplasm. To establish close to native stepping conditions on the microtubules, we lysed native HEK293T cells (further denoted as native lysate), HEK293T cells overexpressing mCherry-TRAK1 (further denoted as TRAK1 lysate) and HEK293T cells overexpressing the purification tag only (further denoted as Halo lysate; Methods). We supplemented these lysates with KIF5BΔ-GFP to visualize its motion along microtubules (Fig. 4d and Supplementary Fig. 4a). The median run length of KIF5BΔ-GFP in native lysate was 0.57 μm (CI$_{95}$ (0.53, 0.62) μm, $n = 529$ molecules), and thereby, as previously reported[28], much shorter than on bare microtubules. Strikingly, in TRAK1 lysate, the median run length increased about twofold to 1.09 μm (CI$_{95}$ (0.99,

1.23) μm, $n = 714$ molecules) (Fig. 4e). Consistently, the median interaction time of KIF5BΔ-GFP increased by 60% from 1.02 s (CI$_{95}$ (1.00, 1.20) s, $n = 529$ molecules) in native lysate to 1.60 s (CI$_{95}$ (1.43, 1.80) s, $n = 714$ molecules) in TRAK1 lysate (Fig. 4f). This increase in the run length and interaction time was not observed in Halo lysate overexpressing the purification tag (Supplementary Fig. 4b, c), demonstrating that TRAK1 enhances the KIF5B processivity on highly crowded microtubules in cell lysate. Taken together, our results demonstrate that TRAK1 promotes long range KIF5BΔ-based transport on crowded microtubules.

**TRAK1 enables mitochondrial transport in vitro.** Finally, we asked if the KIF5B-TRAK1 transport complex characterized above can transport mitochondria in vitro. We thus isolated mitochondria expressing mitochondria-targeted eGFP (denoted as mitochondria-GFP) from 4T1 cells using a mild isolation protocol ("Methods"). We verified by mass spectrometry that these mitochondria did not contain TRAK1, but contained the trans-membrane protein Miro (see Supplementary Data 1), which acts as an adaptor protein for the interaction of mitochondria with the transport complex[40]. We then tested the interaction of mitochondria-GFP with microtubules ("Methods"). No processive movement and only rare stationary binding of mitochondria-GFP to microtubules was observed in absence of both TRAK1 and KIF5BΔ (landing rate $0.87 ± 1.41$ mitochondria mm$^{-1}$ min$^{-1}$ (mean ± standard deviation, $n = 4$ mitochondria, $N = 6$ experiments)) as well as in presence of only KIF5BΔ (landing rate $1.02 ± 1.89$ mitochondria mm$^{-1}$ min$^{-1}$ (mean ± standard deviation, $n = 2$ mitochondria, $N = 6$ experiments)). Interestingly, in absence of KIF5BΔ but in presence of mCherry-TRAK1, stationary as well as diffusive interactions of mCherry-TRAK1-mitochondria-GFP complexes with microtubules were observed and the landing rate increased to $4.18 ± 1.60$ mitochondria mm$^{-1}$ min$^{-1}$ (mean ± standard deviation, $n = 21$ mitochondria, $N = 4$ experiments), showing that TRAK1 can tether mitochondria to microtubules (Supplementary Fig. 5a). When we combined mitochondria-GFP with mCherry-TRAK1 and KIF5BΔ, we observed that mitochondria-GFP colocalized with KIF5BΔ-bound mCherry-TRAK1, associated with microtubules with a landing rate of $3.68 ± 0.46$ mitochondria mm$^{-1}$ min$^{-1}$ (mean ± standard deviation, $n = 10$ mitochondria, $N = 3$ experiments), and subsequently moved processively along the microtubule (Fig. 5a) (Table 1). We also observed processive movement of mCherry-TRAK1 not associated with mitochondria, presumably in complex with unlabelled KIF5BΔ. Mitochondria moved in presence of mCherry-TRAK1 and unlabelled KIF5BΔ processively with an average velocity of $266 ± 148$ nm s$^{-1}$ (mean ± standard deviation, $n = 72$ mitochondria), slower than both KIF5BΔ-GFP ($918 ± 171$ nm s$^{-1}$ (mean ± standard deviation), same dataset as in Fig. 2f) and KIF5BΔ-GFP-mCherry-TRAK1 transport complexes ($599 ± 262$ nm s$^{-1}$ (mean ± standard deviation), same dataset as in Fig. 2f) (Fig. 5b). Consistently, KIF5BΔ-GFP exhibited the shortest median run length of 1.54 μm (CI$_{95}$ (1.43, 1.79) μm, same dataset as in Fig. 2d), which was increased to 5.77 μm (CI$_{95}$ (3.78, 6.87) μm, same dataset as in Fig. 2d) in presence of mCherry-TRAK1 (Fig. 5c). The median run length of mitochondria-GFP was determined to be longer than 20 μm ($n = 72$ mitochondria; Fig. 5c; "Methods"). We estimated on average three to four mCherry-TRAK1 dimers to be bound to a single mitochondrion (Fig, 5d; "Methods"), which can explain the increased run length and decreased velocity of mitochondria-GFP. In summary, we show that supplementing isolated mitochondria with recombinant KIF5BΔ and TRAK1 enables processive mitochondrial motility along microtubules in vitro. Moreover, we show that

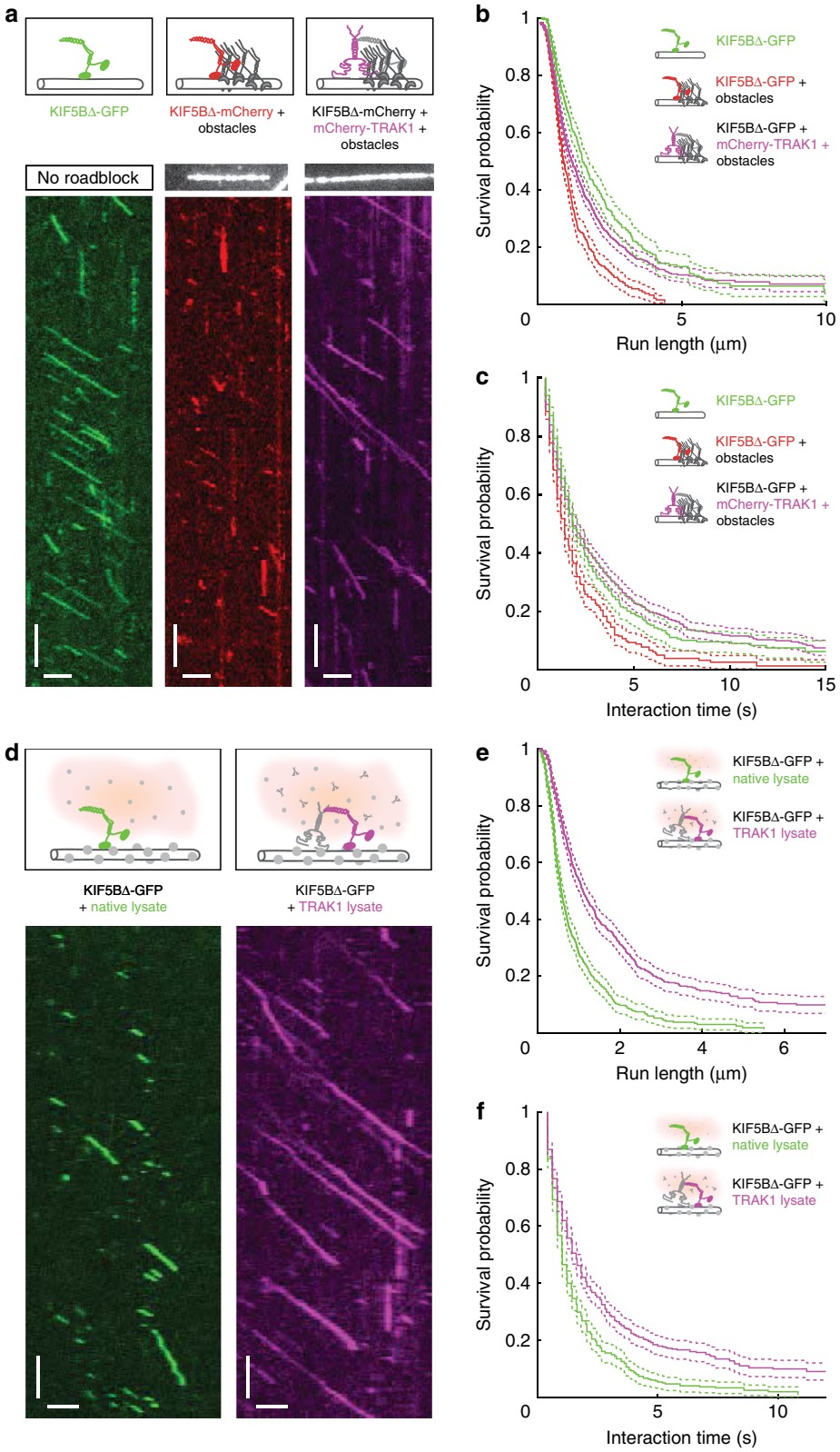

mitochondria travel further but move slower than single KIF5BΔ molecules or KIF5BΔ-TRAK1 transport complexes.

## Discussion

During the processive motion of dimeric kinesin-1 molecular motors the individual motor domains alternately disengage and engage with the microtubule surface. The engaged motor domain provides anchoring for the disengaged motor domain, which is searching for the next binding site on the microtubule. If the anchoring motor domain disengages before the next binding site is found, the molecular motor unbinds from the microtubule, which terminates its run. Therefore, a low detachment rate of the

**Fig. 4 TRAK1 increases KIF5B processivity in crowded environments. a** Schematic illustrations and kymographs of processively moving KIF5BΔ-GFP (green), KIF5BΔ-GFP in presence of obstacles (red) and KIF5BΔ-GFP in presence of mCherry-TRAK1 and obstacles (magenta), respectively. Horizontal scale bars 1 µm, vertical 10 s. **b**, **c** Survival probability of the run length and interaction time of KIF5BΔ-GFP (green, same datasets as in Fig. 2d, e), KIF5BΔ-GFP in presence of obstacles (red, $n = 547$ molecules, $N = 3$ experiments) (both $p$-values $< 10^{-4}$) and KIF5BΔ-GFP in presence of mCherry-TRAK1 and obstacles (magenta, $n = 1066$ molecules, $N = 3$ experiments) (both $p$-values $< 10^{-4}$). The median run length and interaction time decreased for KIF5BΔ-GFP in presence of obstacles but increased in presence of mCherry-TRAK1 despite the presence of obstacles. **d** Schematic illustrations and kymographs of processively moving KIF5BΔ-GFP in native cell lysate (green) and in TRAK1-overexpressing cell lysate (magenta), respectively. Horizontal scale bars 1 µm, vertical 10 s. **e**, **f** Survival probability plots showing the increased median run length and interaction time of KIF5BΔ-GFP in TRAK1 lysate (magenta, $n = 529$ molecules, $N = 6$ experiments) in comparison to native lysate (green, $n = 714$ molecules, $N = 6$ experiments) (both $p$-values $< 10^{-4}$). All two-tailed $p$-values were obtained by a log-rank test. See also Supplementary Fig. 4. Source data are provided as a Source data file.

anchoring motor domain extends the time available for locating the next binding site and increases the processivity of the molecular motor[50]. While binding of TRAK1 to kinesin-1 was reported previously[39,41–43,45,46], we here found that TRAK1, via its C-terminus, also directly interacts with the microtubule, diffusively tethering KIF5B to the microtubule surface. This tethering increases the processivity of the molecular motor, likely by extending the time available for the disengaged motor domain of KIF5B to search for the next free binding site on the microtubule (Fig. 6a). This increase in processivity comes at the cost of decreased velocity, which may be due to a combination of the frictional drag of TRAK1 when moved along the microtubule, conformational changes of the molecular motor or steric hindrance by the interaction of TRAK1 with the molecular motor. Analogously, MAP7D3, a mammalian isoform of MAP7, has been reported to recruit KIF5B to the microtubule, where it increases KIF5B processivity while simultaneously decreasing its velocity[48,51,52]. We hypothesize that the anchoring role of TRAK1 could be further regulated e.g. by TRAK1 post-translational modifications, which were shown to alter mitochondrial trafficking[53].

Microtubule surfaces in cells are crowded by numerous microtubule-associated proteins, which act as obstacles for processively moving molecular motors. When kinesin-1 encounters an obstacle, its run length decreases as kinesin-1 is more likely to dissociate from the microtubule[26,28] than to continue its run by switching the protofilament by side-stepping and circumventing the obstacle[25]. In vitro single molecule experiments in cell lysate mimic the physiological state of microtubules crowded with microtubule-associated proteins[28,54] while rigor-binding kinesin-1 mutants provide stationary obstacles irreversibly masking the kinesin-1 binding sites[25]. Our experiments showed that TRAK1 strongly increased KIF5B processivity under these crowding conditions on the microtubule surface. We suggest that the additional anchoring of KIF5B by TRAK1 increases the time the molecular motor can pause in front of an obstacle without detaching from the microtubule. The extended time available increases the probability that either the obstacle vacates the occupied binding site or that the molecular motor side-steps to circumvent the obstacle.

The intrinsically disordered microtubule-associated protein tau presents a distinctive type of obstacle. In healthy neurons, the tau concentration increases from the cell body to the synapse[55,56] whereas its distribution is reversed in neurodegenerating neurons[57]. Tau is an important regulator of microtubule-based transport, it decreases the run length of plus-end directed cargo transport[29,30] and inhibits kinesin-dependent trafficking of mitochondria[31]. Knockdown of TRAK1 was demonstrated to result in neurodegeneration, which was suppressed by an additional knockdown of tau[44]. In vitro, tau cooperatively forms cohesive islands on the microtubule surface, preventing kinesin-1 stepping within the tau island-coated regions of microtubules[32,34]. Our results show that the combined affinity of

TRAK1 and KIF5B enables the complex to enter regions covered with cohesive tau islands, increasing the probability of KIF5B traversing them and thus overriding the tau island-dependent blockade of kinesin-1-driven transport (schematically shown in Fig. 6b). Mechanistically, we propose that TRAK1-mediated anchoring to the microtubule increases the KIF5B waiting time in front of the tau island's edge, promoting the probability of the KIF5B motor domain to displace the outmost of tau's microtubule-binding repeats when the repeat transiently vacates its binding site. By this mechanism KIF5B can sequentially displace the tau microtubule-binding repeats, enabling motion through the tau island-coated microtubule region. This hypothesis is supported by the lower velocity of the KIF5B-TRAK1 transport complex observed within the islands. Similarly, Kip3, a super-processive kinesin known to pause in absence of an adjacent free binding site[58], has been shown to traverse cohesive tau islands efficiently[32]. Other neuronal intrinsically disordered microtubule-associated proteins are likely to form similar cohesive islands on microtubules[52,59]. We hypothesize that the interaction of TRAK1 or other adaptor proteins with molecular motors could differentially regulate axonal transport by fine-tuning the motility of molecular motors in microtubule regions coated by distinct cohesive islands.

Apart from providing additional anchoring points, TRAK1 enables KIF5B processivity by direct activation of the molecular motor in a kinesin-1 light chain-independent mechanism. In absence of cargo, kinesin-1 is in an inactive conformation caused by the binding of its C-terminal cargo-binding domain to the N-terminal motor domain, which prevents the molecular motor from binding to and processively moving along microtubules[35–37]. Binding of cargo to the tail of kinesin-1 heavy chain can activate processive motility of kinesin-1[36,60]. The Drosophila Milton protein (TRAK ortholog) was found to interact directly with the kinesin-1 heavy chain without requiring kinesin-1 light chains[41,43], and the TRAK2-binding site was mapped to the C-terminus of kinesin-1 KIF5A and KIF5C, respectively[45,46], suggesting that TRAK could activate processive motility of kinesin-1. Here we provide direct evidence for this notion and show that the TRAK1 activation of KIF5B is independent of the TRAK1 interaction with microtubules, indicating an allosteric activation mechanism. To date only few proteins have been shown to activate kinesin-1 through a direct interaction with its heavy chain. Experiments in cell extract revealed Sunday Driver (JIP3) as one of the activators[61], while experiments in minimal cell free systems revealed an activating role for Ran-binding protein 2 (RanBP2)[62] and MAP7D3[51]. Here, using a minimal system, we found that TRAK1 belongs to this small group of proteins directly activating kinesin-1 heavy chain (schematically shown in Fig. 6b).

By reconstituting mitochondrial transport in vitro using recombinant TRAK1 and KIF5B as well as isolated mitochondria, we showed that KIF5B and TRAK1 constitute a minimal transport complex that can drive directed mitochondrial motion along microtubules (schematically shown in Fig. 6b) and that TRAK1

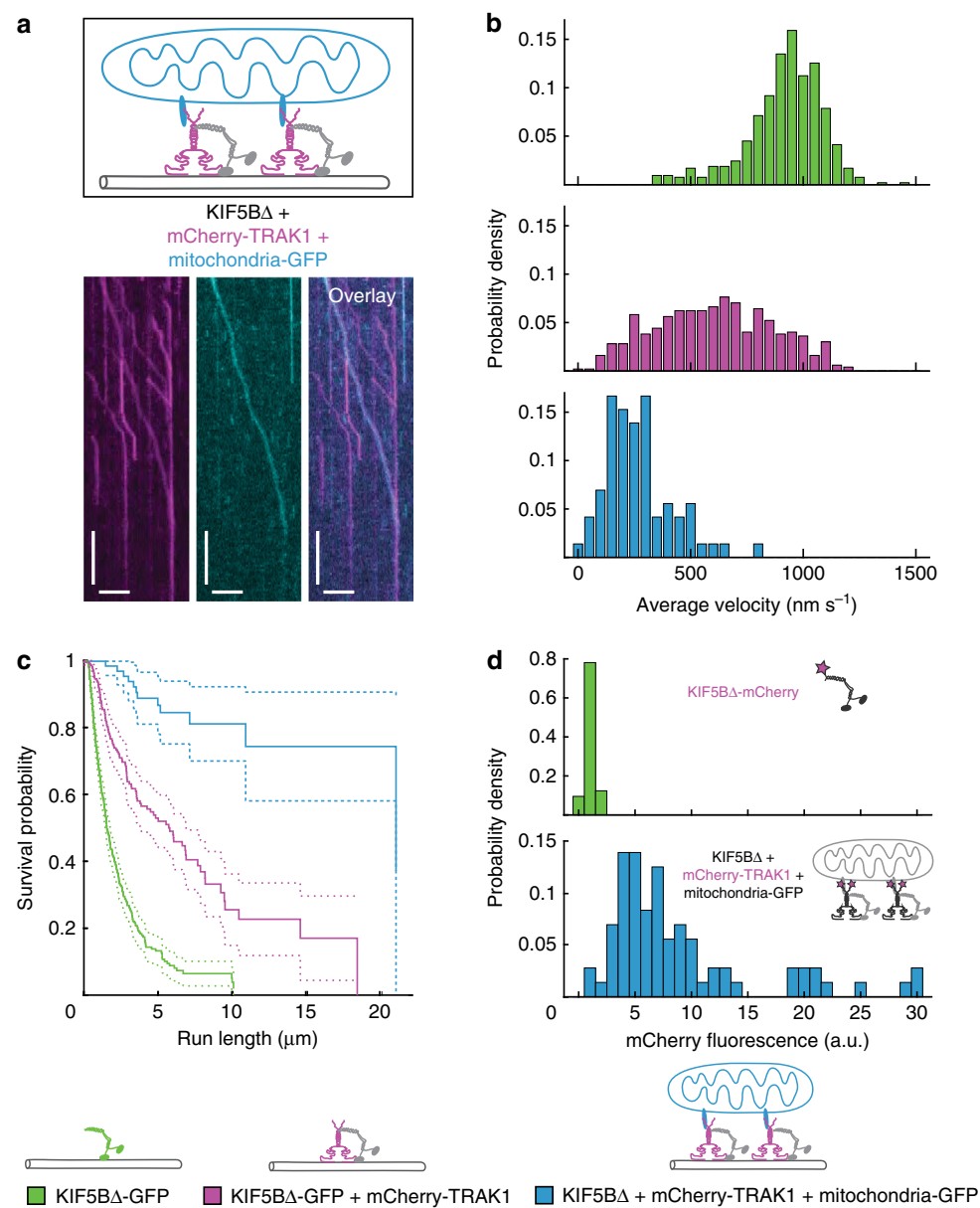

**Fig. 5 TRAK1 enables mitochondrial transport in vitro. a** Schematic illustration and kymographs of mitochondria-GFP (cyan) being processively transported along microtubules by mCherry-TRAK1 (magenta), bound to unlabelled KIF5BΔ. Horizontal scale bars 5 μm, vertical 10 s. See also Supplementary Fig. 5. **b** Histograms of the average velocities of the processive movement of KIF5BΔ-GFP (green, same dataset as in Fig. 2f), decreased in presence of mCherry-TRAK1 (magenta, same dataset as in Fig. 2f) and further decreased for mitochondria-GFP transported by KIF5BΔ-mCherry-TRAK1 complexes (cyan, $n = 72$ mitochondria, $N = 10$ experiments). **c** Survival probability plot showing an increased median run length of mitochondria-GFP transported by KIF5BΔ-mCherry-TRAK1 complexes (cyan, $n = 72$ mitochondria, $N = 10$ experiments) in comparison to KIF5BΔ-GFP (green, same dataset as in Fig. 2d) ($p < 10^{-4}$) and KIF5BΔ-GFP-mCherry-TRAK1 complexes (magenta, same dataset as in Fig. 2d) ($p < 10^{-4}$). Two-tailed $p$-values were obtained by a log-rank test. **d** Histograms of the background subtracted fluorescence intensity of KIF5BΔ-mCherry (green, 1.0 ± 0.5 (median ± interquartile range (IQR), $n = 105$ molecules, $N = 2$ experiments)) and of mCherry-TRAK1 in presence of KIF5BΔ and mitochondria-GFP (cyan, 6.7 ± 5.77 (median ± IQR, $n = 73$ mitochondria, $N = 10$ experiments)) showing on average three to four mCherry-TRAK1 dimers bound to a single mitochondrion (Methods). For labelling efficiencies see Supplementary Table 1. For a list of proteins present after the crude isolation of mitochondria refer to the Supplementary Data 1 (Mass Spectrometry of mitochondria). Source data are provided as a Source data file.

**Table 1 Binding behaviour of mitochondria to microtubules and the respective landing rates.**

| Condition | Type of microtubule interaction | Landing rate [number mm$^{-1}$ min$^{-1}$] |
|---|---|---|
| Mitochondria-GFP | Sporadic stationary binding | 0.87 ± 1.41 |
| Mitochondria-GFP + KIF5BΔ | Sporadic stationary binding | 1.02 ± 1.89 |
| Mitochondria-GFP + mCherry-TRAK1 | Diffusion | 4.18 ± 1.60 |
| Mitochondria-GFP + KIF5BΔ + mCherry-TRAK1 | Directional motility | 3.68 ± 0.46 |

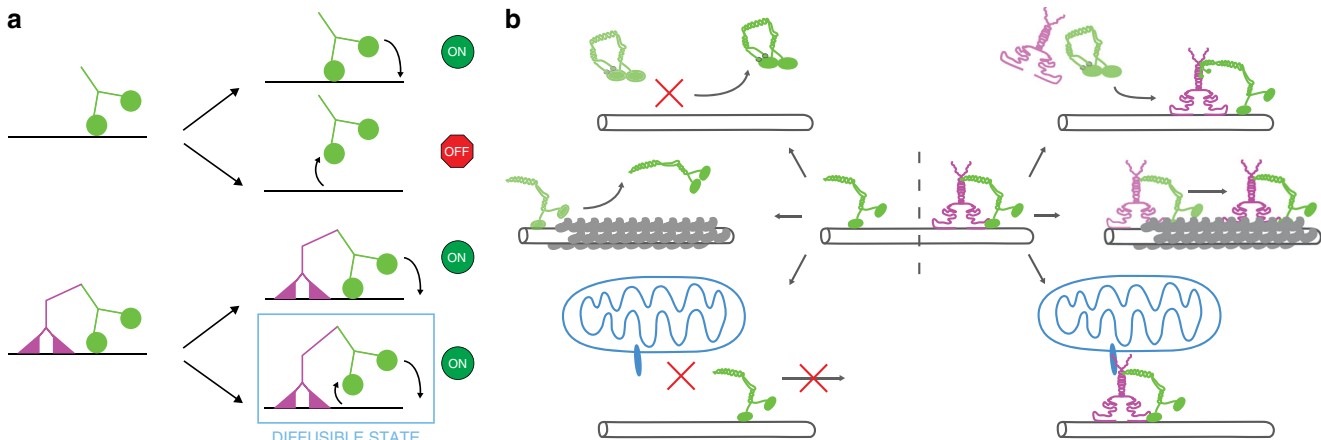

**Fig. 6 Schematic illustration of the TRAK1-mediated anchoring of KIF5B. a** Top: in absence of TRAK1, KIF5B (green) can either continue its walk by rebinding the disengaged motor domain to the microtubule or dissociate from the microtubule when the engaged motor domain unbinds from the microtubule. Bottom: in presence of microtubule-bound TRAK1 (magenta), when both motor domains of KIF5B disengage from the microtubule, KIF5B remains tethered to the microtubule through a diffusive interaction of TRAK1 with the microtubule and thereby enables the rebinding of a motor domain of KIF5B to the microtubule. In this state, TRAK1 might facilitate navigation around obstacles by diffusion along the microtubule surface. **b** Overview of the functions of TRAK1. Top: TRAK1 activates auto-inhibited KIF5B, enabling its processive movement along microtubules. Middle: TRAK1 increases the processivity of KIF5B in crowded environments. Bottom: TRAK1 enables KIF5B-based transport of isolated mitochondria along microtubules in vitro.

can tether mitochondria to microtubules in absence of KIF5B. The run length of transported mitochondria increased in comparison to KIF5B and KIF5B-TRAK1, respectively, presumably due to the presence of multiple TRAK1-KIF5B transport complexes, in line with previous studies on multi-motor transport[17–20]. Though we estimated on average three to four mCherry-TRAK1 dimers bound to the mitochondria, the percentage of the TRAK1 molecules engaged in the transport might vary, since likely not all detected TRAK1 molecules form a complex with KIF5B and not all KIF5B-TRAK1 complexes may be positioned such that they interact with the microtubule. The run length of mitochondria in axons of cultured hippocampal neurons is in the order of tens of micrometres[15] and the anterograde velocity can range from 0.1 to 0.8 $\mu$m s$^{-1}$[42] to 1 $\mu$m s$^{-1}$ estimated in mice[63], comparable to values obtained in our experiments. Our in vitro reconstituted system, assembled using a minimal number of components, provides a powerful tool to test the mechanisms proposed to underpin the motion of mitochondria and to explore the regulatory roles of the individual components of the mitochondria trafficking machinery.

In summary, we demonstrate that TRAK1 anchors kinesin-1 to the microtubule surface, increasing the processivity of kinesin-1, to enable microtubule-based transport in the crowded cytoplasmic environment. We propose tethering by auxiliary proteins as a general mechanism regulating molecular motors and other filament-associated proteins.

## Methods

**Production of recombinant proteins**. Full-length mCherry-TRAK1 was cloned using Gateway Cloning. The cDNA encoding TRAK1 with an N-terminal mCherry-tag termed KIAA1042 (accession number Q9UPV9-1) was obtained from the Kazusa DNA Research Institute (Japan). The mCherry-TRAK1-encoding nucleotide sequences were PCR amplified using specifically designed primer pairs (Supplementary Table 2). For the generation of the gateway entry clone, the nucleotide sequence was inserted by means of a BP recombinant reaction (Invitrogen, Thermo Fisher Scientific, Carlsbad, CA, USA) according to the manufacturer's protocol into a pDONR221 donor vector. The entry clone was verified by Sanger sequencing prior to transferring the entry clone into the destination vector. For the generation of the expression plasmid, an LR recombinant reaction was performed, generating a destination vector containing a TEV-cleavage site and a TwinStrep-FLAG-Halo-tag at the N-terminus of TRAK1. All TRAK1 constructs were expressed in HEK293T cells (a kind gift from Radu A. Aricescu, University of Oxford, Oxford, England), grown in Free Style F17 medium (Gibco, Thermo Fisher Scientific, Inc., Waltham, MA,

USA) supplemented with 0.1% Pluronic F-68 (Invitrogen, Thermo Fisher Scientific, Carlsbad, CA, USA) and 2 mM L-glutamine at 110 rpm under a humidified 5% CO$_2$ atmosphere at 37 °C. In all, 1 mg ml$^{-1}$ linear polyethylene imine (Polysciences Inc., Warrington, PA, USA) and 0.7 mg of the expression plasmid were incubated in phosphate buffered saline (PBS) for 10 min prior to the addition to 350 ml cells at the concentration of $4 \times 10^6$ cell ml$^{-1}$. Four hours post transfection, the cells were diluted twofold in ExCell serum-free medium. Four days post transfection, the cells were harvested by centrifugation at 4 °C for 10 min at 500 × g. The cell pellet was resuspended in ice-cold lysis buffer (100 mM Tris-HCl, 10 mM NaCl, 5 mM KCl, 2 mM MgCl$_2$, 10% glycerol, pH 8.0) supplemented with benzonase (1 U ml$^{-1}$; Merck, Darmstadt, Germany) and a protease inhibitor cocktail (cOmplete, EDTA free, Roche, Basel, Switzerland) and lysed by pulsed sonication for 5 min (20 s pulses with 24 W min$^{-1}$). Cell lysis was further assisted by the addition of Igepal-630 to the final concentration of 0.2% (v/v) during 20 min incubation on ice with occasional mixing. NaCl was added to the final concentration of 150 mM and the mixture was further incubated on ice for 20 min. Insoluble material was removed by centrifugation at 9.000 × g for 15 min at 4 °C. The cell lysate was further cleared by a second centrifugation step at 30,000 × g for 30 min at 4 °C. The supernatant was loaded onto a StrepTactinXT column (IBA, Gottingen, Germany) equilibrated in lysis buffer with 150 mM NaCl for affinity chromatography. After washing the column with wash buffer (100 mM Tris-HCl, 150 mM NaCl, 1 mM EDTA, pH 8), the protein was eluted by cleaving off the N-terminal tag with 1:20 (w/w) TEV protease in the wash buffer overnight at 4 °C. The next day, the eluted protein was collected, concentrated using an Amicon ultracentrifuge filter with a molecular weight cutoff of 100 kDa (Merck, Darmstadt, Germany) and loaded onto a Superose 6 10/300 GL column (GE Healthcare Bio-Sciences, Little Chalfont, UK) for further separation by size exclusion chromatography with 100 mM Tris-HCl pH 8.0, 150 mM NaCl, 2 mM MgCl$_2$, 1 mM EDTA, 0.1% tween, 10% glycerol, 1 mM DTT, 0.1 mM ATP as a mobile phase. The purified protein was concentrated using an Amicon ultracentrifuge filter and flash frozen in liquid nitrogen.

The mCherry-TRAK1 deletion mutant mCherry-TRAK1Δ was obtained by inserting a stop codon after amino acid 635 of the mCherry-TRAK1 encoding nucleotide sequence by means of a PCR-based mutagenesis (Agilent Technologies, QuikChange II Site-Directed Mutagenesis Kit) according to the manufacturers protocol (for primers see Supplementary Table 2). For the expression and purification of mCherry-TRAK1Δ the protocol as described above was followed.

KIF5B constructs were obtained by PCR amplification of the cDNA (from Dharmacon, Lafayette, CO, USA, GeneBank accession number BC126281) of the amino acids 1-963 for full-length KIF5B and 1-905 for KIF5BΔ using primers containing AscI- and NotI-digestion sites flagging the particular KIF5B-encoding nucleotide sequences (Supplementary Table 2). After AscI-NotI-digestion of the inserts, they were ligated into an AscI-NotI-digested FlexiBAC destination vector containing a C-terminal fluorescent tag (GFP or mCherry) followed by a 3C PreScission protease cleavage site and a 6xHis-tag. All KIF5B constructs were expressed in SF9 insect cells (Expression systems, Davis, CA, USA) using the opensource FlexiBAC baculovirus vector system for protein expression[64]. The insect cells were harvested after 4 days by centrifugation at 300 × g for 10 min at 4 °C in an Avanti J-26S ultracentrifuge (JLA-9.1000 rotor, Beckman Coulter, Brea, CA, USA). The cell pellet was resuspended in 5 ml ice-cold PBS and stored at −80 °C for further use. For cell lysis, the insect cells were homogenized in 30 ml ice-cold His-

Trap buffer (50 mM Na-phosphate buffer, pH 7.5, 5% glycerol, 300 mM KCl, 1 mM MgCl$_2$, 0.1% tween-20, 10 mM BME, 0.1 mM ATP) supplemented with 30 mM imidazole, protease inhibitor cocktail and benzonase to the final concentration of 25 units ml$^{-1}$, and centrifuged at 45,000 × g for 60 min at 4 °C in the Avanti J-26S ultracentrifuge (JA-30.50Ti rotor, Beckman Coulter, Brea, CA). The cleared cell lysate was incubated for 2 h at 4 °C with a lysis buffer-equilibrated Ni-NTA column (HisPur Ni-NTA Superflow Agarose, Thermo Scientific, Thermo Fisher Scientific, Inc., Waltham, MA, USA) on a rotator for subsequent affinity chromatography via the C-terminal 6xHis-tag. The Ni-NTA column was washed with wash buffer (His-Trap buffer supplemented with 60 mM imidazole) and the protein was eluted with elution buffer (His-Trap buffer supplemented with 300 mM imidazole). The fractions containing the protein of interest were pooled, diluted 1:10 in the His-Trap buffer and the purification tag was cleaved overnight by 3C PreScisson protease. The solution was reloaded onto a Ni-NTA column to further separate the cleaved protein from the 6xHis-tag. The protein was concentrated using an Amicon ultracentrifuge filter and flash frozen in liquid nitrogen.

The expression plasmid for the obstacle-kinesin was an eGFP-labelled rigor binding kinesin-1 mutant from *Rattus norvegicus*, which contains the N-terminal 430 amino acids with a point mutation of amino acid 93 from threonine to asparagine and with a C-terminal eGFP- and 6xHis-tag[25]. It was expressed in *Escherichia coli* strain BL21(DE3) and purified via affinity chromatography using a Ni-NTA column as described above. The final cleavage of the 6xHis-tag was omitted.

The human tau isoform htau441[65] with a C-terminal 6xHis-tag and a mCherry- or GFP-tag, respectively, was expressed in SF9 insect cells and purified by affinity chromatography using the 6xHis-tag as described above.

**Stoichiometry estimation**. To estimate the stoichiometry of TRAK1 vs. KIF5B molecules in the transport complex, we first estimated the average number of active mCherry fluorophores on constitutively dimeric KIF5BΔ-mCherry and on mCherry-TRAK1 (i.e. the labeling efficiencies) by measuring the mCherry- and protein-absorptions in size exclusion chromatography. Using the respective extinction coefficients, this estimation yielded labeling efficiencies of about 22% for KIF5BΔ-mCherry and about 88% for mCherry-TRAK1 (Supplementary Table 1). This indicates that a large majority of KIF5BΔ-mCherry dimers contained only one active mCherry-fluorophore, while a large majority of mCherry-TRAK1 dimers contained two mCherry-fluorophores. We later compared the fluorescence intensity distributions of KIF5BΔ-mCherry with mCherry-TRAK1 in the motility experiments.

**Microtubules**. Unlabeled and fluorescently labeled (80% unlabeled and 20% Alexa Fluor 647 NHS ester-labeled; Invitrogen, Thermo Fisher Scientific, Carlsbad, CA, USA) microtubules were polymerized from 4 mg ml$^{-1}$ porcine tubulin for 2 h at 37 °C in BRB80 (80 mM PIPES, 1 mM EGTA, 1 mM MgCl$_2$, pH 6.9) supplemented with 1 mM MgCl$_2$ and 1 mM GMPCPP (Jena Bioscience, Jena, Germany). The polymerized microtubules were centrifuged for 30 min at 18,000 × g in a Microfuge 18 Centrifuge (Beckman Coulter, Brea, CA) and the pellet was resuspended in BRB80 supplemented with 10 μM taxol (BRB80T). For microtubules used in experiments involving cohesive tau islands, a polymerization mixture of 25% DMSO, 20 mM MgCl$_2$ and 5 mM GTP in BRB80 was prepared on ice and 1.25 μl of the mixture was added to 5 μl of 4 mg ml$^{-1}$ porcine tubulin. Microtubules were polymerized for 30 min at 37 °C. Subsequently, 100 μl BRB80T was added prior to centrifugation and resuspension as described above.

**Preparation of cell lysates for microscopy**. Cell lysates of untransfected cells (native lysate) and cells transfected with DNA encoding TwinStrep-FLAG-Halo-mCherry-TRAK1 (TRAK1 lysate) or TwinStrep-FLAG-Halo-GFP (Halo lysate) were prepared from HEK293T cells. The cells were harvested by centrifugation for 5 min at 500 × g at 4 °C and the cell pellets were resuspended in 0.5 pellet volumes of lysis buffer (12 mM K-PIPES at pH 6.8, 1 mM MgCl$_2$, 1 mM EGTA supplemented with 10 μg ml$^{-1}$ cytochalasin D and a protease inhibitor cocktail), followed by pulsed sonication. Insoluble material was removed by centrifugation for 30 min at 20,000 × g at 4 °C. With the addition of 10 μg ml$^{-1}$ cytochalasin D the polymerization of actin filaments was prevented. The cell lysate was used directly or was flash frozen in liquid nitrogen and stored at −80 °C for further use.

**Isolation of mitochondria**. Murine mammary carcinoma cell line 4T1 stably expressing mitochondria-targeted eGFP was prepared by transfection with the pTagGFP2-mito vector (Evrogen, Moscow, Russia) using Lipofectamine 3000 (Thermo Scientific, Thermo Fisher Scientific, Inc., Waltham, MA, USA), followed by clonal selection with G418 (Sigma Aldrich, St. Louis, Missouri, USA). Cells were broken by a Balch-style homogenizer (Isobiotec, Heidelberg, Germany) at 8 μm clearance. Mitochondria were then isolated by centrifugation for 8 min at 800 × g at 4 °C, followed by pre-clearance of the supernatant for 5 min at 3000 × g at 4 °C and a final collection of the mitochondrial pellet by centrifugation for 15 min at 10,000 × g at 4 °C. The mitochondrial pellet was washed three times with isolation buffer (250 mM sucrose, 1 mM EDTA, 10 mM Tris/Mops pH 7.6) and resuspended in the same buffer, and the protein concentration was determined by a BCA assay (Thermo Scientific, Thermo Fisher Scientific, Inc., Waltham, MA, USA).

Mitochondria were kept on ice and used within 4 h in the microscopy experiments. Mitochondria isolated on three different days were analyzed by mass spectrometry to exclude the presence of co-purified TRAK1.

**In vitro motility assay**. For the construction of flow cells, glass coverslips (22 × 22 mm$^2$ and 18 × 18 mm$^2$; Corning, Inc., Corning, NY) were cleaned in piranha solution (H$_2$O$_2$/H$_2$SO$_4$) prior to silanization with 0.05% dichlorodimethylsilane (DDS) in trichloroethylene. Next, two DDS coated coverslips of different sizes were glued together using heated strips of parafilm M (Pechiney Plastic Packaging, Chicago, IL) and the flow cell was incubated with 10 μg ml$^{-1}$ anti-tubulin antibody (Sigma Aldrich, T7816) in PBS for 10 min prior to surface passivation by 1% Pluronic F-127 in PBS. After at least 60 min, the flow cell was washed using BRB80T prior to immobilization of microtubules. Unbound microtubules were removed after 2 min by BRB80T and subsequently the buffer was exchanged by motility buffer (BRB80 containing 10 μM taxol, 10 mM dithiothreitol, 20 mM D-glucose, 0.1% Tween-20, 0.5 mg ml$^{-1}$ casein, 1 mM Mg-ATP, 0.22 mg ml$^{-1}$ glucose oxidase and 20 μg ml$^{-1}$ catalase). For experiments on bare microtubules, KIF5B constructs were diluted such that single molecules interacting with microtubules could be visualized (6 nM KIF5B-GFP, 0.2 nM KIF5BΔ-GFP or 1.3 nM KIF5BΔ-mCherry, respectively), or 175 nM TRAK1 constructs were flushed in motility buffer directly into the flow cell, or a mixture of the two proteins was pre-incubated 10 min on ice prior to flushing into the flow cell for the imaging of KIF5B-TRAK1 complexes of various constructs. For decorating microtubules with obstacles, 0.1 nM rigor-binding kinesin-1 mutant in the motility buffer were flushed into the flow cell with immobilized microtubules. After an incubation for several seconds, unbound obstacles were removed using motility buffer. Subsequently, 2 nM KIF5BΔ-GFP or a pre-incubated complex of 2 nM KIF5BΔ-GFP and 100 nM mCherry-TRAK1, was flushed into the flow cell. For decorating microtubules with tau islands, 3.5 nM tau-mCherry or tau-GFP in motility buffer was flushed into the flow cell containing immobilized microtubules. Tau islands formed during the incubation time of up to 5 min and unbound tau was removed using the motility buffer. Subsequently, 2 nM KIF5BΔ-GFP and/or 100 nM mCherry-TRAK1 in motility buffer was added to the flow cell while keeping the tau concentration in the solution constant. Experiments in cell lysates were performed by incubation of the cell lysates for 20 min on ice in an oxygen scavenger (20 mM glucose, 160 μg ml$^{-1}$ glucose oxidase and 20 μg ml$^{-1}$ catalase) followed by the addition of 2 mM Mg-ATP and 0.2 nM KIF5BΔ-GFP prior to flushing into the flow cell[28]. Mitochondrial transport was reconstituted by pre-incubating 2 nM unlabeled KIF5BΔ, 100 nM mCherry-TRAK1 and 10 μg mitochondria-GFP on ice prior to flushing the mixture into the flow cell with immobilized microtubules. All experiments were performed at room temperature. To estimate the characteristic photobleaching times of GFP and mCherry, microtubules were densely decorated with high concentrations of KIF5B-GFP (60 nM) and KIF5B-mCherry (13 nM) in motility buffer as described above but Mg-ATP was substituted by 1 mM AMPPNP to immobilize the molecular motor on the microtubule surface. Unbound motor proteins were removed using the AMPPNP-containing motility buffer prior to imaging.

**Fluorescence microscopy and image acquisition**. For fluorescence imaging, the total internal reflection fluorescence (TIRF) mode of an inverted widefield fluorescence microscope (Nikon Eclipse Ti-E; Nikon, Tokyo, Japan) equipped with a motorized XY stage and a perfect focus system was used together with a 60× oil immersion objective (Nikon CFI Apo TIRF 60x Oil, NA 1.49, WD 0.12 mm), a 2.5× relay lens in front of an electron-multiplied charge-coupled device camera (iXon Ultra DU-888; Andor, Belfast, Northern Ireland) and, if necessary, an additional 1.5× magnifying tube lens. Alexa647-labelled microtubules, mCherry- and GFP-labelled proteins and GFP-labelled mitochondria were visualized by the sequential switching between a Cy5 filter (632-652, 669-741), TRITC filter (556-566, 593-668) and FITC filter (483-493, 500-550) or by using a Quad Band Set filter (405/488/561/640). The position of unlabeled microtubules and mitochondria were determined using an interference reflection microscopy unit. Fluorescence images were acquired for one to two minutes with 200 ms exposure time and 300 gain multiplier using NIS-Elements Advanced Research software v5.02 (Laboratory Imaging). Experiments were performed over several months, each experiment presented was repeated at least on three individual days. No data were excluded from the study.

**Fluorescence image analysis**. Image analysis for estimating the motility parameters (interaction time, run length and velocity in Figs. 1, 2, and 4 and Supplementary Figs. 2 and 4) as well as the intensity distributions (Supplementary Fig. 2) of fluorescently labelled KIF5BΔ and TRAK1 was performed by tracking the movement of the respective molecules with the high-precision tracking software FIESTA[66]. All trajectories were double-checked by eye to avoid computer misinterpretations. The following trajectories were not analyzed: stationary molecules not exhibiting processive movements, clustering molecules, molecules passing crossing microtubules and stationary molecules accumulating at the microtubule plus-end. For determining the survival probability in terms of Kaplan–Meier estimations, the following trajectories were denoted as censored: trajectories reaching the end of a microtubule, trajectories starting in the first or ending in the last imaging frame, trajectories starting or ending at the edge of the field of view. Further trajectory evaluation was performed in MatLab

using traces consisting of at least three data points (The MathWorks, Natick, MA, USA). The interaction time was determined as the time difference between the beginning and the end of the trajectory and the run length as the distance along the path of the trajectory. Kaplan–Meier estimations were evaluated using the MatLab build-in empirical cumulative distribution function (ecdf), which computes the 95% confidence interval using the Greenwood's formula. Two-tailed $p$-values were determined by a log-rank test and scattered graphs were plotted using the UnivarScatter script. Calculated velocities represent the average velocity of a molecule from the beginning to the end of its processive movement along a microtubule. Frame-to-frame velocities (Supplementary Fig. 2i) were computed by extracting the position of a molecule in each frame and calculating the velocities between two consecutive frames. The run length, interaction time and velocity of molecules within tau-mCherry islands were determined by manually measuring the time and position of the molecule when entering the tau island and when leaving the tau-mCherry island by either traversing the whole island or dissociating from the microtubule, respectively, using the image processing software Fiji[67] (Fig. 3). The lengths of tau-mCherry islands were as well measured manually using Fiji (Supplementary Fig. 3). Molecules that did not enter tau-mCherry islands were not included in this evaluation. Single mitochondria were identified using the criteria of round shape in interference reflection microscopy (Supplementary Fig. 5b) and fluorescence microscopy and a diameter of around 1 μm. The velocity and run length of mitochondria (Fig. 5b, c) were measured manually by determining the position at the beginning and the end of the movement using the Fiji software. Fluorescence intensities of the mCherry signal of mitochondria transported processively by KIF5BΔ-mCherry-TRAK1 complexes and of KIF5BΔ-mCherry (Fig. 5d) were measured in the beginning of their interaction with the microtubule in a square of $13 \times 13$ pixel. The fluorescence intensity of the background, measured in an adjacent square of $13 \times 13$ pixel, was subtracted. Time traces of fluorescent intensities (Supplementary Fig. 2c–e) were determined by a line scan along the trace of the molecule of interest in the respective kymograph. Landing rates (Fig. 1a and Table 1) were calculated by counting the number of molecules or mitochondria, respectively, per unit length of microtubule and per unit time.

**Photobleaching effects estimation**. To estimate characteristic photobleaching times of GFP and mCherry, images were acquired continuously for 40 min using the imaging conditions described above. Characteristic photobleaching times of GFP and mCherry were estimated by fitting a single exponential curve to the intensity signal over time yielding 250 s for GFP and 167 s for mCherry. As these decay times are order of magnitude higher than the measured interaction times of KIF5B and TRAK1, we neglected the effect of photobleaching in our experiments.

**Reporting summary**. Further information on research design is available in the Nature Research Reporting Summary linked to this article.

## Data availability
Data supporting the findings of this manuscript are available from the corresponding authors upon reasonable request. A reporting summary for this Article is available as a Supplementary Information file. The source data underlying Figs. 1a, 2a–f, 3d–h, 4b, c, 4e, f and 5b–d and Supplementary Figs. 1b–c, 2c–i, 3 and 4b, c are provided as a Source data file and is together with Supplementary Data 1 available in figshare with the identifier https://doi.org/10.6084/m9.figshare.12311867.

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

## Acknowledgements

We thank the Protein Facility of MPI-CBG Dresden, Yulia Bobrova and Karel Harant from the Laboratory of Mass Spectrometry at BIOCEV for technical support. We acknowledge the financial support from GAUK (grant no. 250440 to V.H.), the Czech Science Foundation (grant no. 18-08304S to Z.L., 20-04068S to M.B., 19-20553S, 20-18513S to J.R. and 18-10832S to J.N.), the Australian Research Council Discovery (grant no. DP180103426 to J.N.), the Introduction of New Research Methods to BIOCEV (CZ.1.05/2.1.00/19.0390) project from the ERDF, the institutional support from the CAS (RVO: 86652036), the MS facility of CMS supported by MEYS CR (LM2015043), and the Imaging Methods Core Facility at BIOCEV, an institution supported by the MEYS CR (Large RI Project LM2018129 Czech-BioImaging) and ERDF (project no. CZ.02.1.01/0.0/0.0/16_013/0001775) for their support in obtaining imaging data presented in this paper.

## Author contributions

Conceptualization, J.R., M.B., Z.L.; Methodology, V.H., J.R., C.B., M.B., Z.L.; formal analysis, V.H.; investigation, V.H.; resources, V.H., Z.N., L.G., S.D., J.N.; Writing—original draft, V.H.; review and editing, V.H., J.N., S.D., J.R., M.B., Z.L.; visualization, V.H.; supervision, J.R., M.B., Z.L.; funding acquisition, V.H., J.R., J.N., M.B., Z.L.

## Competing interests

The authors declare no competing interests.
