## [Peer Review File · Nature Communications]

Reviewers' comments first round:

Reviewer #1 (Remarks to the Author):

The authors investigate the activation of kinesin-1 by the TRAK1 cargo adapter molecule. These partners are known to interact and there are interesting disease phenotypes of TRAK1 and tau, but the mechanism underlying the TRAK-kinesin interaction are not clear. The authors clearly show that the partners bind, and that TRAK1 enhances the kinesin run-length through TRAK1 tethering the motor to the microtubule. They go on to show that TRAK1 enhances kinesin's ability to traverse islands of tau and use purified mitochondria to show that TRAK1 and kinesin-1 cooperate in multi-motor transport of mitochondria.

The single-molecule data are very solid and the analysis is correct. The work together makes a nice and pretty straightforward story that is relevant for kinesin-based intracellular cargo transport. I had a few comments to strengthen the manuscript, detailed below.

Comments

Top p6. "1.84 s (CI95 (1.64, 2.05) s, n = 534 molecules) in the absence of mCherry-TRAK1 to 12.20 s (CI95 (8.54, 15.86) s, n = 221 molecules) in its presence (Figure 3f)." This is less than an order of magnitude

Page 6. TRAK1 slows kinesin-1 down from 918 to 694 nm/s, which is significant. The authors postulate that this slowing is due to frictional drag from TRAK1. This point should be argued more rigorously, as follows. In Fig 1, the authors track the diffusion of isolated TRAK1. From this, they should be able to estimate the diffusion constant of TRAK1, and from there to measure the drag coefficient of TRAK1 along microtubules. Based on published force-velocity relationships of kinesin-1, is the observed slowing consistent with the predicted drag coefficient?

Page 11: "...it inhibits kinesin-dependent trafficking of mitochondria and thereby implicates Alzheimer's disease". This is a confusing sentence, "implicates" is not the right word here.

The authors bring up some interesting points in the discussion related to tau and TRAK1 interactions. I suggest the authors bring this point up in the introduction to set the stage better. And in general in the introduction, defining the questions more specifically, (rather than just seeing what it does) could significantly strengthen the impact of the results. For instance questions about the tail requirement, the activity on the mitochondria, and other points where experiments were performed but the questions were not introduced in the intro beforehand.

Reviewer #2 (Remarks to the Author):

With their work, the authors wanted to understand how kinesin-1 can function as efficient cargo transporter in crowded environments. The authors report that the kinesin-1-cargo adaptor TRAK1 activates autoinhibited KIF5B and increases the processivity of KIF5B lacking its autoinhibiting C-terminus. This effect is also detectable when microtubules are crowded with Tau protein islands both when using a pure in vitro assay and when imaging in extracts. Finally, the authors report that the KIF5Bdelta-TRAK1 complex transports mitochondria. TRAK1 reduces velocity and increases processivity – an effect that is amplified by the addition of mitochondria. Interestingly, when adding only TRAK1, the effects on the KIF5B motility parameters don't seem to be caused by motor-bundling, but by the microtubule-binding activity of TRAK1 which was discovered in this work.

I think the paper is an important step forward towards understanding full neuronal cargo transport complexes. A lot of these complexes are proposed based on classical cell biology approaches but are then difficult or impossible to build with pure components which indicates that it is not understood how these complexes are actually assembled and which factors are essential.

Major points:

- TRAK1 and KIF5B were made in different expression systems at different temperatures which can affect fluorophore folding and maturation rate. For the fluorescence comparisons in Fig.2d, it would be important to understand, which fraction of the fluorophores are active. Could the authors measure the fluorescence intensities per amount of protein to test this?
- If one TRAK1 dimer binds one KIF5 dimer, I would expect that two TRAK1 dimers would bind two KIF5 dimers and hence TRAK1 should cause KIF5 tetramerization to some extent. How do the authors explain that this is not observed? Or can TRAK1 bind a KIF5B dimer only as tetramer?
- In Fig.3f, the interaction time of KIF5Bdelta-GFP and KIF5Bdelta-GFP+mCherry-TRAK1 complexes is compared and the mCherry-TRAK1 signal is used to report on KIF5 dwell time in the presence of TRAK1. Depending on imaging settings, GFP can bleach faster than mCherry which would artificially reduce the KIF5-GFP dwell times. It would be assuring to provide GFP and mCherry bleaching rates at the chosen imaging conditions.
- The authors state that in control experiments lacking TRAK1 no processive movement of mitochondria is detectable (lines 284-286) but I could not find data supporting this statement. I think it is an important point because it shows that TRAK1 is sufficient for mitochondria transport by KIF5 in vitro. When KIF5Bdelta-mCherry alone is incubated with mitochondria-GFP - are no runs detectable? Or substantially less than with TRAK1?
- The authors speculate that the lower velocity and higher processivity caused by adding mitochondria stems from the recruitment of multiple TRAK1-KIF5 complexes to one mitochondrion, but no data is provided to support this idea. It seems straightforward to provide intensity data supporting this idea.

Minor points:

- Is TRAK1 also a tetramer when it binds to microtubules without KIF5 or is KIF5 interaction triggering TRAK1 tetramerization?
- As TRAK1 can bind microtubules – can it recruit mitochondria to microtubules or are both competing for TRAK1 interaction?
- As TRAK1 promotes KIF5 movement through crowded microtubule regions, and TRAK1-microtubule interactions were identified as the mechanism by which TRAK1 supports KIF5 motility, one would expect that the TRAK1-microtubule interaction is less affected by Tau compared to the KIF5-microtubule interaction. It would be very interesting to see if and to which extent TRAK1-microtubule interactions change in the presence of Tau.
- In line 287 the authors write "single-mitochondria". How is it shown that we are looking at single mitochondria?

Reviewer #3 (Remarks to the Author):

In their manuscript, Henrichs et al. utilize in vitro reconstitution with purified proteins to show that the mitochondrial adaptor protein, TRAK1/Milton, directly interacts and activates kinesin-1 while also tethering it to the microtubule during stepping to increase its processivity both on naked microtubules as well as microtubules crowded with other "obstacles" such as the microtubule-associated protein, tau. They then purify mitochondria from cultured cells and find that this cargo can be robustly transported in vitro by the kinesin-1-TRAK1 complex. Overall, this is a very nice story and is the first study to show a bona fide activator of full-length kinesin-1 in vitro, making it highly significant to a broad audience. If the authors address the below points, it will be acceptable for publication in Nature Communications.

Major Points:

1) They use a TRAK protein construct that lack its C-terminal microtubule-binding domain and find that it cannot tether or increase the processivity of the truncated, active kinesin-1. However, this experiment should be performed with full-length kinesin-1. If TRAK does indeed release the autoinhibition of full-length kinesin-1, it becomes an interesting question of whether TRAK does this allosterically or if it requires its microtubule-binding domain. There are two different mechanisms potentially at play, which do not need to be mutually exclusive, but should be examined more thoroughly. It could be that TRAK has two modes of action on kinesin-1: 1) it

could allosterically release autoinhibition and 2) it could tether an active kinesin-1 motor as it moves processively along the lattice. The C-terminus of TRAK is obviously required for #2, but is it dispensable for #1?

2) To date, TRAK is the first protein that can directly activate kinesin-1 using purified components in an in vitro system. This is incredibly exciting, but the authors should place this in the context of the field to highlight this novelty. For example, many proteins have been reported to activate kinesin-1 in vivo or using lysate experiments in vitro, but none of these proteins have been shown to directly activate kinesin-1 using purified components. In addition, the authors should reference the MAP7 work from my lab (Monroy et al., 2018), the Akhmanova lab (Hooikaas et al., 2019) and the Hendricks lab (Chaudhury et al., 2019) to highlight the contrast. MAP7 directly binds and recruits kinesin-1 to the microtubule, dramatically increasing the number of engaged motors, but does not activate it like TRAK does. Therefore, this is a unique and novel function of TRAK, and placing in in the context of other proposed adaptors will strengthen the significance.

3) For the tau experiments, the authors hypothesize in the discussion that TRAK tethers and enables kinesin-1 to remain bound longer to the microtubule in order to displace tau from the protofilament so that it can move through a dense tau patch. This is interesting because they previously observed that a yeast kinesin can displace tau, and we've also seen that a non-enzymatic MAP, MAP7, can do the same over a longer timescale (Monroy et al., 2018). Can the authors quantify the velocity of kinesin-1 before and after entry into a tau patch? It appears in their figure that the velocity may indeed slow down and this would provide more support for their hypothesis.

4) The authors hypothesize on page 9 that "several" kinesin-TRAK complexes might be attached to a single mitochondrion, but the kymograph in Figure 5 shows a dimmer fluorescence intensity profile of the TRAK associated with the mitochondrion. Can the authors analyze the intensity of the TRAK in these runs to provide an estimate for the number of motors on the cargo? I realize that, based on Figure 1, there are most likely complexes that are not labeled, but if many kymographs are analyzed, it can still provide a general idea of motor number. Or, the authors could correlate run length with fluorescence intensity and compare the same three conditions as in Fig 5c.

Minor Point:

References need to be organized—some include the author/year while others are referenced with numbers (see second paragraph of the introduction as an example).

Signed, Cassandra Ori-McKenney

We thank the reviewers for their appreciation of our work and their constructive comments. We have been able to address their comments and questions and altered the manuscript accordingly. The major changes to the text are marked in blue in the revised manuscript. Please find the detailed replies to the reviewers' comments (in blue) below.

reviewers' comments:

Reviewer #1 (Remarks to the Author):

The authors investigate the activation of kinesin-1 by the TRAK1 cargo adapter molecule. These partners are known to interact and there are interesting disease phenotypes of TRAK1 and tau, but the mechanism underlying the TRAK-kinesin interaction are not clear. The authors clearly show that the partners bind, and that TRAK1 enhances the kinesin run-length through TRAK1 tethering the motor to the microtubule. They go on to show that TRAK1 enhances kinesin's ability to traverse islands of tau and use purified mitochondria to show that TRAK1 and kinesin-1 cooperate in multi-motor transport of mitochondria.

The single-molecule data are very solid and the analysis is correct. The work together makes a nice and pretty straightforward story that is relevant for kinesin-based intracellular cargo transport. I had a few comments to strengthen the manuscript, detailed below.

Comments

Top p6. "1.84 s (CI95 (1.64, 2.05) s, n = 534 molecules) in the absence of mCherry-TRAK1 to 12.20 s (CI95 (8.54, 15.86) s, n = 221 molecules) in its presence (Figure 3f)." This is less than an order of magnitude

We altered the text according to the reviewer's comment.

Page 6. TRAK1 slows kinesin-1 down from 918 to 694 nm/s, which is significant. The authors postulate that this slowing is due to frictional drag from TRAK1. This point should be argued more rigorously, as follows. In Fig 1, the authors track the diffusion of isolated TRAK1. From this, they should be able to estimate the diffusion constant of TRAK1, and from there to measure the drag coefficient of TRAK1 along microtubules. Based on published force-velocity relationships of kinesin-1, is the observed slowing consistent with the predicted drag coefficient?

Based on the diffusion coefficient, we estimated the drag coefficient of TRAK1 and, given the typical velocity at which the TRAK1 molecule is dragged by kinesin-1 along the microtubule surface, we estimated also the frictional force caused by TRAK1. However, based on the published force-velocity relationship, this force is orders of magnitude too low to account for the slowdown of the motor. We thus hypothesize that the slowdown might be caused by a combination of frictional drag, conformational changes in the motor and steric hindrance caused by the interaction of TRAK1 with the motor. We now discuss these possibilities in the text.

Page 11: "...it inhibits kinesin-dependent trafficking of mitochondria and thereby implicates Alzheimer's disease". This is a confusing sentence, "implicates" is not the right word here.

We rephrased the text.

The authors bring up some interesting points in the discussion related to tau and TRAK1 interactions. I suggest the authors bring this point up in the introduction to set the stage better. And in general in the introduction, defining the questions more specifically, (rather than just seeing what it does) could significantly strengthen the impact of the results. For instance questions about the tail requirement, the activity on the mitochondria, and other points where experiments were performed but the questions were not introduced in the intro beforehand.

We thank the reviewer for pointing this out. We have now altered the introduction section according to the reviewer's suggestion.

Reviewer #2 (Remarks to the Author):

With their work, the authors wanted to understand how kinesin-1 can function as efficient cargo transporter in crowded environments. The authors report that the kinesin-1-cargo adaptor TRAK1 activates autoinhibited KIF5B and increases the processivity of KIF5B lacking its autoinhibiting C-terminus. This effect is also detectable when microtubules are crowded with Tau protein islands both when using a pure in vitro assay and when imaging in extracts. Finally, the authors report that the KIF5Bdelta-TRAK1 complex transports mitochondria. TRAK1 reduces velocity and increases processivity – an effect that is amplified by the addition of mitochondria. Interestingly, when adding only TRAK1, the effects on the KIF5B motility parameters don't seem to be caused by motor-bundling, but by the microtubule-binding activity of TRAK1 which was discovered in this work.

I think the paper is an important step forward towards understanding full neuronal cargo transport complexes. A lot of these complexes are proposed based on classical cell biology approaches but are then difficult or impossible to build with pure components which indicates that it is not understood how these complexes are actually assembled and which factors are essential.

Major points:

- TRAK1 and KIF5B were made in different expression systems at different temperatures which can affect fluorophore folding and maturation rate. For the fluorescence comparisons in Fig.2d, it would be important to understand, which fraction of the fluorophores are active. Could the authors measure the fluorescence intensities per amount of protein to test this?

This is an important point, which we overlooked. Many thanks for pointing this out. We now estimated the amount of active fluorophore per protein (labeling efficiency) in size exclusion chromatography using the mCherry absorbance and protein absorbance. This estimate shows that about 90% of the TRAK1 molecules, but only about 20% of the KIF5B molecules are mCherry-labeled. The KIF5B-mCherry intensity peak thus corresponds to a single mCherry signal of a single-mCherry-labeled dimer, which will be detected with way higher probability than double-mCherry-labeled dimer (probability of 0.32 compared to 0.04). The mCherry-TRAK1 peak, which is centered at twice higher intensity, thus represents the double-mCherry-labeled dimer of TRAK1. We performed new analysis and conclude now that a dimer of KIF5B interacts with a dimer of TRAK1. We altered the main text, the methods and Figure S2d-j.

- If one TRAK1 dimer binds one KIF5 dimer, I would expect that two TRAK1 dimers would bind two KIF5 dimers and hence TRAK1 should cause KIF5 tetramerization to some extent. How do the authors explain that this is not observed? Or can TRAK1 bind a KIF5B dimer only as tetramer?

As discussed above, we now performed new analysis and conclude that the KIF5B-TRAK1 complex comprises one KIF5B dimer and one TRAK1 dimer (Figure S2d-j). We changed the text accordingly.

- In Fig.3f, the interaction time of KIF5Bdelta-GFP and KIF5Bdelta-GFP+mCherry-TRAK1 complexes is compared and the mCherry-TRAK1 signal is used to report on KIF5 dwell time in the presence of TRAK1. Depending on imaging settings, GFP can bleach faster than mCherry which would artificially reduce the KIF5-GFP dwell times. It would be assuring to provide GFP and mCherry bleaching rates at the chosen imaging conditions.

To estimate the bleaching rates of GFP and mCherry we performed control experiments using KIF5BΔ-GFP and KIF5BΔ-mCherry bound to microtubules in presence of AMPPNP. These experimental conditions assure that the motor proteins are bound to the microtubules stationary, without unbinding, diffusion or directed motion. We imaged the motors using a 488 nm and 561 nm laser, respectively, using the settings as in the experiments. The proteins bleached with a characteristic time of 250 s for GFP and of 167 s for mCherry. These times are at least an order of magnitude higher than the measured interaction times of 1.84 s for KIF5BΔ-GFP and 12.20 s for KIF5BΔ-GFP-mCherry-TRAK1 complexes. The effect of photobleaching is thus negligible. We now mention these control experiment in the Methods section.

- The authors state that in control experiments lacking TRAK1 no processive movement of mitochondria is detectable (lines 284-286) but I could not find data supporting this statement. I think it is an important point because it shows that TRAK1 is sufficient for mitochondria transport by KIF5 in vitro. When KIF5Bdelta-mCherry alone is incubated with mitochondria-GFP - are no runs detectable? Or substantially less than with TRAK1?

We performed additional control experiments and analyzed the landing rate and motility pattern of mitochondria-GFP. As indicated in the manuscript, Miro, the mitochondria-anchored binding partner of TRAK1, was bound to the mitochondrial membrane after isolation as confirmed by Mass Spectrometry. The following experimental situations were now analyzed: 1) only mitochondria-GFP, 2) mitochondria-GFP in presence of mCherry-TRAK1, 3) mitochondria-GFP in presence of KIF5BΔ and 4) mitochondria-GFP in presence of KIF5BΔ and mCherry-TRAK1. We observed directional motility only in presence of both, KIF5BΔ and mCherry-TRAK1. In the presence of only mCherry-TRAK1 we observed static binding and diffusion of mitochondria-GFP along microtubules. In the presence of only KIF5BΔ or in the absence of both proteins, we observed sporadic static binding of mitochondria-GFP to the microtubules. We added these new results to Figure 5b and S5a.

- The authors speculate that the lower velocity and higher processivity caused by adding mitochondria stems from the recruitment of multiple TRAK1-KIF5 complexes to one mitochondrion, but no data is provided to support this idea. It seems straightforward to provide intensity data supporting this idea.

To determine the number of KIF5 Δ -mCherry-TRAK1 complexes on mitochondria-GFP, we evaluated their mCherry-fluorescence intensities and compared it with the fluorescence intensities of single processively moving KIF5 Δ -mCherry molecules. While the intensity distributions indicate the presence of in average three to four mCherry-TRAK1 dimers on the surface of mitochondria-GFP, we cannot elucidate the exact number of TRAK1-molecules involved in the transport of mitochondria mostly because i) a fraction of the mitochondria-localized mCherry-TRAK1 molecules might not be attached to KIF5 Δ and thereby do not participate in their transport and/or ii) a fraction of the mCherry-TRAK1 molecules might be bound to the mitochondria in an area that is not in contact with the microtubule and thereby are not able to participate in the transport. We added histograms of the background subtracted fluorescence intensities of KIF5 Δ -mCherry and of mCherry-TRAK1 in presence of KIF5 Δ and mitochondria-GFP to Figure 5e and we discuss these results.

Minor points:

- Is TRAK1 also a tetramer when it binds to microtubules without KIF5 or is KIF5 interaction triggering TRAK1 tetramerization?

As mentioned above, we now re-analysed our data and found that TRAK1 interacts with kinesin1 as a single dimer. We now also measured the fluorescence intensity distribution of mCherry-TRAK1 diffusing along microtubules in absence of KIF5 Δ and the results show that also in this situation TRAK1 is present as a dimer. We added this new data in the results section and to Figure S2d and S2g.

- As TRAK1 can bind microtubules – can it recruit mitochondria to microtubules or are both competing for TRAK1 interaction?

New control experiments with mitochondria in the presence of TRAK1 but absence of KIF5 Δ showed that TRAK1 can recruit mitochondria for stationary binding and diffusion to microtubules. The landing rate of TRAK1-mitochondria complexes was about five times higher than of mitochondria themselves. We now added these new findings to the Figure 5b and added a supplementary figure S5a showing representative kymographs of mitochondria-GFP in presence of only of mCherry-TRAK1 interacting with microtubules.

- As TRAK1 promotes KIF5 movement through crowded microtubule regions, and TRAK1-microtubule interactions were identified as the mechanism by which TRAK1 supports KIF5 motility, one would expect that the TRAK1-microtubule interaction is less affected by Tau compared to the KIF5-microtubule interaction. It would be very interesting to see if and to which extent TRAK1-microtubule interactions change in the presence of Tau.

We performed single molecule experiments with TRAK1 in the presence of tau island-coated microtubules to determine the TRAK1-microtubule interaction under these conditions. We observed that TRAK1 is in most cases restricted to areas outside the tau islands. The TRAK1-microtubule interaction does not seem to be strong enough for TRAK1 to penetrate a tau island. It seems that only the combined affinity of the KIF5B-TRAK1 transport complex is high enough to allow the complex to enter into the tau islands. We now include these new results in the manuscript in Figure 3a.

- In line 287 the authors write “single-mitochondria”. How is it shown that we are looking at single mitochondria?

Before imaging the fluorescence signal of KIF5B-mCherry-TRAK1-mitochondria-GFP, we always imaged the situation using interference reflection microscopy. This allowed us to directly see the mitochondria and to determine whether they appeared as single ones or whether it was a cluster of several mitochondria (the latter occurring mainly after long imaging times, which we did not include in the analysis) - assuming that single mitochondria are around 1 μm in diameter and are round. We added a supplementary figure S5b showing different representative interference reflection microscopy images of mitochondria and described the procedure in the methods.

Reviewer #3 (Remarks to the Author):

In their manuscript, Henrichs et al. utilize in vitro reconstitution with purified proteins to show that the mitochondrial adaptor protein, TRAK1/Milton, directly interacts and activates kinesin-1 while also tethering it to the microtubule during stepping to increase its processivity both on naked microtubules as well as microtubules crowded with other “obstacles” such as the microtubule-associated protein, tau. They then purify mitochondria from cultured cells and find that this cargo can be robustly transported in vitro by the kinesin-1-TRAK1 complex. Overall, this is a very nice story and is the first study to show a bona fide activator of full-length kinesin-1 in vitro, making it highly significant to a broad audience. If the authors address the below points, it will be acceptable for publication in Nature Communications.

Major Points:

- 1) They use a TRAK protein construct that lack its C-terminal microtubule-binding domain and find that it cannot tether or increase the processivity of the truncated, active kinesin-1. However, this experiment should be performed with full-length kinesin-1. If TRAK does indeed release the autoinhibition of full-length kinesin-1, it becomes an interesting question of whether TRAK does this allosterically or if it requires its microtubule-binding domain. There are two different mechanisms potentially at play, which do not need to be mutually exclusive, but should be examined more thoroughly. It could be that TRAK has two modes of action on kinesin-1: 1) it could allosterically release autoinhibition and 2) it could tether an active kinesin-1 motor as it moves processively along the lattice. The C-terminus of TRAK is obviously required for #2, but is it dispensable for #1?

Thanks for this great suggestion. To determine the mode of action of TRAK1 on full length KIF5B, we now directly compared its activation by TRAK1 and TRAK1Δ. With both TRAK1 constructs, the number of activated KIF5B molecules was comparable. Next, we analyzed the run length, interaction time and velocity of the activated KIF5B molecules. We measured a longer run length and interaction time but lower velocity in the presence of TRAK1 in comparison to TRAK1Δ. This indicates that the tethering of KIF5B to the microtubules through TRAK1 is not essential for the activation of KIF5B, but is essential for increasing the motor processivity. We now added the comparison of the activation of KIF5B by TRAK1 and TRAK1Δ by means of: 1) landing rate, 2) run length, 3) interaction time and 4) average velocity to Figure 1a and 2a-c and we discuss the allosteric mechanism of kinesin-1 activation by TRAK1.

2) To date, TRAK is the first protein that can directly activate kinesin-1 using purified components in an in vitro system. This is incredibly exciting, but the authors should place this in the context of the field to highlight this novelty. For example, many proteins have been reported to activate kinesin-1 in vivo or using lysate experiments in vitro, but none of these proteins have been shown to directly activate kinesin-1 using purified components. In addition, the authors should reference the MAP7 work from my lab (Monroy et al., 2018), the Akhmanova lab (Hooikaas et al., 2019) and the Hendricks lab (Chaudhury et al., 2019) to highlight the contrast. MAP7 directly binds and recruits kinesin-1 to the microtubule, dramatically increasing the number of engaged motors, but does not activate it like TRAK does. Therefore, this is a unique and novel function of TRAK, and placing in in the context of other proposed adaptors will strengthen the significance.

Thanks for pointing this out, we now bring our findings more in context in the discussion and we discuss the mentioned references.

3) For the tau experiments, the authors hypothesize in the discussion that TRAK tethers and enables kinesin-1 to remain bound longer to the microtubule in order to displace tau from the protofilament so that it can move through a dense tau patch. This is interesting because they previously observed that a yeast kinesin can displace tau, and we've also seen that a non-enzymatic MAP, MAP7, can do the same over a longer timescale (Monroy et al., 2018). Can the authors quantify the velocity of kinesin-1 before and after entry into a tau patch? It appears in their figure that the velocity may indeed slow down and this would provide more support for their hypothesis.

We now analyzed the average velocities of KIF5BΔ in presence of TRAK1 within tau islands and in non-island regions. Indeed, the transport complex slows down within the island. We hypothesize that the transport complex spends more time waiting for the next binding site while traversing an island. We added a graph to Figure 3f comparing the average velocity of KIF5BΔ in presence of TRAK1 within tau islands and in non-island regions and discuss this new result.

4) The authors hypothesize on page 9 that “several” kinesin-TRAK complexes might be attached to a single mitochondrion, but the kymograph in Figure 5 shows a dimmer fluorescence intensity profile of the TRAK associated with the mitochondrion. Can the authors analyze the intensity of the TRAK in these runs to provide an estimate for the number

of motors on the cargo? I realize that, based on Figure 1, there are most likely complexes that are not labeled, but if many kymographs are analyzed, it can still provide a general idea of motor number. Or, the authors could correlate run length with fluorescence intensity and compare the same three conditions as in Fig 5c.

We have now quantified the number of TRAK1 molecules on the mitochondria using the TRAK1 signal. However, as discussed above (Reviewer 2, point no 5), we do not know if all these TRAK1 molecules are bound to kinesin-1 and/or bound to the microtubule, and thus engaged in driving the transport. We added the histograms of the background subtracted fluorescence intensities of mCherry-TRAK1 present on the motile mitochondria-GFP to Figure 5 and we discussed these results.

Minor Point:

References need to be organized—some include the author/year while others are referenced with numbers (see second paragraph of the introduction as an example).

We now organized all the references according the Nature Communications style.

Signed, Kassandra Ori-McKenney

Thank you, Kassandra, your comments are much appreciated.

Peer Review File

Reviewers' comments second round:

Reviewer #2 (Remarks to the Author):

The authors have addressed all concerns adequately and I have no objections to the publication of this work.

Reviewer #3 (Remarks to the Author):

The authors have addressed my concerns either entirely or to the best of their ability, and I support publication of this study in Nature Communications. The new data with the TRAKdeltaC that lacks the microtubule binding domain is incredibly exciting. The dual mode of allosteric activation and tethering of kinesin-1 opens up many exciting avenues of future research. My only other suggestion would be to move certain data from the supplement, such as Figure S1c-e, to the main figures, but this is at the discretion of the authors.

RESPONSE TO REVIEWERS' COMMENTS:

Reviewer #2 (Remarks to the Author): The authors have addressed all concerns adequately and I have no objections to the publication of this work.

Reviewer #3 (Remarks to the Author): The authors have addressed my concerns either entirely or to the best of their ability, and I support publication of this study in Nature Communications. The new data with the TRAKdeltaC that lacks the microtubule binding domain is incredibly exciting. The dual mode of allosteric activation and tethering of kinesin-1 opens up many exciting avenues of future research. My only other suggestion would be to move certain data from the supplement, such as Figure S1c-e, to the main figures, but this is at the discretion of the authors.

Response:

We now rearranged the figures and moved the suggested data from Supplementary Figure 1 to Figure 1.